

# Projected changes in forest fire season, number of fires and burnt area in Fennoscandia by 2100

Outi Kinnunen[1], Leif Backman[1], Juha Aalto[2,3], Tuula Aalto[1], and Tiina Markkanen[1]

[1]Climate Research, Finnish Meteorological Institute, Erik Palménin aukio 1, 00560 Helsinki, Finland.

[2]Weather and climate change impact research, Finnish Meteorological Institute, Erik Palménin aukio 1, 00560 Helsinki, Finland.

[3]Department of Geosciences and Geography, University of Helsinki, P.O. Box 64, Gustaf Hällströmin katu 2a, 00014 Helsinki, Finland

**Correspondence:** Outi Kinnunen (outi.kinnunen@fmi.fi)

**Abstract.** Forest fire dynamics are expected to alter due to climate change. Despite the projected increase in precipitation, rising temperatures will amplify forest fire risk from the present to the end of the century. Here, we analysed the changes in fire season, number of fires and burnt area in Fennoscandia from 1951 to 2100. The JSBACH-SPITFIRE ecosystem model regional simulations were done under two climate change forcing scenarios (RCP 4.5 and RCP 8.5) and three global climate driver models (CanESM2, CNRM-CM5 and MIROC5) with a 0.5 ° resolution. Simulations were forced by downscaled and bias-corrected EURO-CORDEX data. Generally, as a consequence of the projected longer fire season and drier fuel, the probability of fires is projected to increase. However, changes in fire season, number of fires and burnt area are very dependent on the climate projection and location; the fire season is estimated to increase by (20–52) days on average, starting (10–23) days earlier and ending (10–30) days later, from the reference period (1981–2010) to the end of the century (2071–2100). The results for Finland indicate a (-7–98) % change in the number of fires and a (-19–87) % change in the burnt area. These findings contribute to a better understanding of potential changes in the future fire seasons in Northern Europe.

| Climate scenarios | JSBACH-SPITFIRE model | Changes in burnt area from 1981–2010 to 2071–2100 in Fennoscandia |
|---|---|---|
| Increase of temperature and precipitation | Fuel properties<br>Ignition rate<br>Fire danger index<br>Fire season<br>Number of fires<br>Burnt area<br>$CO_2$ flux | |



# 1 Introduction

Forest fires are natural to boreal ecosystems. Historically, forest fires are one of the primary disturbances in Fennoscandia,
(Engelmark, 1999; Ramberg et al., 2018; Tolonen and Pitkänen, 2004), where the forests are dominated by Scots pine and
Norway spruce. Nowadays, due to efficient fire monitoring and suppression, fire is a relatively rare event in Fennoscandia
(Ramberg, 2020; Vajda et al., 2014). For example, on average, only 0.006 % (less than 2000 ha) of Swedish forests burn per
year, although in the exceptionally dry summer of 2018, there was a high number of forest fires (Ramberg et al., 2018). In
Finland, the annual burnt area is 500–600 ha (Aalto and Venäläinen, 2021). Newer the less Lehtonen et al. (2016) point out
that the number of large forest fires (> 10 ha) may even triple during the present century. Furthermore, in the Siberian Arctic,
an almost exponential increase has been observed in burnt areas (Descals et al., 2022). Today, population density, landscape
patterns, prescribed fires, and fire prevention policies strongly influence forest fires (Flannigan et al., 2009; Larjavaara et al.,
2005a). The only natural cause of fires is lightning (Larjavaara et al., 2005b; Gromtsev, 2002), and fires are mainly ignited,
about 90 % in Finland (Mäkelä, 2015; Kilpeläinen et al., 2010), by human actions (Flannigan et al., 2009; Ramberg, 2020;
Hantson et al., 2015). The population density is low in Finland, and fires are more likely to occur in the more densely populated
southern part than in the northern part of the country (Vajda et al., 2014; Larjavaara et al., 2005a).

Forest fires have a multitude of impacts on the ecosystem and climate. Fires cause tree mortality, release nutrients (Kulmala
et al., 2014) and create open space in the forest, which helps vegetation to grow from seeds (Venevsky et al., 2002; Moritz et al.,
2012; Ramberg, 2020). Prescribed and reoccurring fires are needed to increase structural heterogeneity within a landscape and
are pivotal for maintaining high biodiversity (Kuuluvainen, 2002; Ramberg, 2020). Mostly, total ecosystem recovery from
severe forest fires is possible, but it will take a long time. Recovery of the forest floor mass could take several years, and
for organic horizons, up to 20 years (Mäkipää et al., 2023). Fire may heat the soil to high temperatures and cause carbon
and nutrient losses, soil physical alterations, stability of organic matter, and mortality of faunal and microbial communities
(Mäkipää et al., 2023). Burning may also have a negative impact on carbon storage due to the losses of C and N in the
ecosystem during the fire (Mäkipää et al., 2023; Mäkelä, 2015). Forest fires are a source of black carbon that is harmful to the
health and accelerates the melting of snow and ice in the Arctic (Aalto and Venäläinen, 2021). In addition, forest fires may
release pollutants, such as mercury (Hg), into the atmosphere, which may cause health problems (Turetsky et al., 2006). Fires
cause greenhouse gas emissions (e.g. carbon dioxide $CO_2$) to be released into the atmosphere, which can further propagate
climate change.

Forest fires are a result of interactions between climate, weather, fuel and humans (Flannigan et al., 2009; Lasslop et al.,
2018). Warmer temperatures have a threefold link to wildfires due to increased evapotranspiration, lightning activity, and fire
season (Flannigan et al., 2009). Climate change influences the spatial variations and the potential of extreme conditions of fire
weather. Fire weather is defined as the weather variables (temperature, precipitation, humidity and wind) that influence fire
behaviour, ignition and suppression (Flannigan et al., 2009). The expected rate of temperature increase due to climate change
in Fennoscandia is 2–3 times higher than the global average (Kaplan and New, 2006; Venäläinen et al., 2020). This expected
acceleration increases the risk of forest fires, although an increase in precipitation might partially have the opposite effect





(Mäkelä, 2015; Aakala et al., 2018; Kilpeläinen et al., 2010; Venäläinen et al., 2020). Across the boreal region, the burnt area and fire occurrences are projected to increase (Flannigan et al., 2009; de Groot et al., 2013) mainly due to temperature increase, but it is modulated by changes in precipitation, litter production, soil respiration and population density. Understanding the
factors behind changes in forest fire activity is essential to ensure effective management of spreading fires (Jolly et al., 2015).

The main focus of previous studies has been the impact of varying meteorological conditions on fire risk (Flannigan et al., 2009; Ramberg, 2020; Mäkelä, 2015). Here, we use the land surface model JSBACH to study the fire season, number of fires and burnt area under changing climate. In this study, we investigate how and why projected climate change modulates the forest fire risk in Fennoscandia. We present results from an analysis of climate change impacts on fire season, the number of
fires and burnt area based on the JSBACH-SPITFIRE model simulations that were forced by downscaled and bias-corrected EURO-CORDEX data. The simulations were made under two climate change scenarios (RCP 4.5 and RCP 8.5) and three global climate driver models (CNRM-CM5, MIROC5 and CanESM2) from 1951 to 2100.

## 2 MATERIALS AND METHODS

### 2.1 Ecosystem model JSBACH

The JSBACH ecosystem model (Kaminski et al., 2013) was developed as the land surface component of the Earth System Models of the Max-Planck Institute for Meteorology (Reick et al., 2021; Mauritsen et al., 2019). The SPITFIRE (SPread and InTensity of FIRE) is a mechanistic global fire model (Thonicke et al., 2010) that has been implemented in the JSBACH ecosystem model (Lasslop et al., 2014). The forest fires in the JSBACH-SPITFIRE ecosystem model are disturbances that depend on weather conditions, fuel properties and population density.

The amount of fuel is estimated from the Yasso07 above-ground C-pools (Goll et al., 2015). It is simulated as a balance between litter produced by the vegetation and soil carbon decomposition, including combustion and consequent dead wood input from fire events. The fuel is divided into fuel classes according to reaction time to atmospheric conditions: 1 h, 10 h, 100 h and 1000 h fuel. The division represents different sizes, i.e. the different surface-area-to-volume rations, of the fuel elements such as leaves, branches and trunks (Lasslop et al., 2014). The time required to reach the equilibrium moisture content under
defined atmospheric conditions is longer with larger fuel elements due to a lower surface-area-to-volume ratio than for fine fuel elements. Grass (live fuel) is included in the 1 h fuel class. After a fire, the burnt carbon is subtracted from the C-pools and released in to the atmosphere as $CO_2$ and carbon from tree mortality is added to the C-pools (Thonicke et al., 2010).

The moisture content of the fuel is exponentially dependent on the Nesterov index (NI), and NI is weighted by the relative amounts of three fuel classes (1h – 100 h). The NI describes the drying power and depends on both temperature and precipi-
tation. The NI (Onderka and Melicherčik, 2010) is a cumulative function of daily maximum and dew point temperature. The index is summed over days when the daily precipitation is less than 3 mm and the dew point temperature is above zero degrees (Thonicke et al., 2010; Running et al., 1987). In JSBACH-SPITFIRE, the fire danger index (FDI) is the probability that an ignition event will cause a spreading fire. The FDI is one for completely dry fuel and zero for insufficient or wet fuel. The FDI is calculated from environmental dryness, temperature and the availability of fuel (Reick et al., 2021). We used the FDI to



estimate the number of days with high fire risk. An FDI > 0.8 indicates very high or extremely high fire risk (Thonicke et al., 2010). The fire season is the period when the FDI is above zero, and forest fires are possible. We defined the length of the fire season as the number of days between the first and last day when the FDI is greater than zero.

In the SPITFIRE, the fire may start from lightning ignition or a human act. The total ignition rate is the sum of lightning and human-caused ignitions. The expected number of human-caused ignition events depends on the population density and the

propensity of people to cause ignition events (Thonicke et al., 2010), which reflect regional and cultural differences (Lasslop and Kloster, 2017). The number of human-caused ignition events is a non-linear function of population density. The ignition events increase with population density until it starts to decline due to landscape fragmentation, urbanisation and associated infrastructural changes (Thonicke et al., 2010). When there is fuel available, and the fuel is dry enough, the ignition will lead to a spreading fire. The number of fires per area is obtained by multiplying the number of total ignition events by the FDI.

The burnt area is determined based on the number of fires, fire duration and the rate of spread (Rothermel, 1972), assuming an elliptical spread pattern (Carmody, 1992). The analysed burnt area is calculated taking into account the woody plant functional types.

## 2.2  Regional simulations

Regional simulations were performed using the JSBACH-SPITFIRE ecosystem model over the period 1951–2100. The simu-

lations were forced by downscaled and bias-corrected data from the EURO-CORDEX initiative (Jacob et al., 2014). We used data on the Eur-44 domain (Earth System Grid Federation Data Node), which was re-gridded to a regular 0.5° lat-lon grid, using nearest neighbour interpolation. The simulated model domain (Fig. 1 a) was limited to the land area within 55–71° N and 5–34° E.

The global models were forced under two Representative Concentration Pathway (RCP) scenarios: RCP 4.5 and RCP 8.5,

with the number indicating radiative forcing values in W/m2 in 2100 (van Vuuren et al., 2011). The RCP 4.5 scenario represents intermediate and RCP 8.5 high greenhouse gas emissions. The RCPs share a common GHG pathway up to 2005 (the historical period) and deviate from there on (scenario period). The regional climate model RCA4 (Samuelsson et al., 2011) was used as a downscaling model for all three global climate driving models (CNRM-CM5, MIROC5 and CanESM2), and a distribution-based bias-correction method (SMHI-DBS45-EOBS12-1981-2010) was applied for all data sets that we used e.g. Yang et al.

(2010). The daily bias-corrected data of precipitation and temperature for both RCP 4.5 and RCP 8.5 were used. In addition, daily data for relative humidity, wind speed and longwave and shortwave radiation were used.

A spin-up of the soil carbon pools was made before the actual simulation using driver data randomly generated from 1951–1980 and a prescribed $CO_2$ concentration (285 ppm). The JSBACH model was run with a timestep of 30 minutes, and values for the variables were output with a daily frequency. The land cover in the simulations represents the current land cover

and was derived from the Finnish CORINE and the European CORINE (European Environment Agency, 2020) data. The ESA LandUse-CCI (European Space Agency, 2019) data was used for the area not covered by CORINE data. The land cover classes were translated into the 11 plant functional types that the JSBACH model uses. Land cover changes were not accounted for.



Soil properties were set according to Hagemann and Stacke (2015) so that the peat fraction of the land area was set according to the map by Xu et al. (2017), while the parameter values of Loamy Sand were assumed for the remaining land area.

The human-caused ignition events were calculated from population density. The historical population density was based on data from the History Database of the Global Environment (Klein Goldewijk et al., 2017). The future scenario follows a middle-of-the-road shared socioeconomic pathway (SSP2) (Jones and O'Neill, 2016). The lightning ignition rate was obtained from a climatology for Northern Europe compiled by the Finnish Meteorological Institute (Mäkelä et al., 2014) that was based on observations from lightning location sensors. The LIS/ODT 0.5 Degree High Resolution Monthly Climatology was used

east of 32E (Cecil, 2016). The LIS/ODT climatology reports total flashes, but only about 20 % are cloud to ground flashes, which was taken into account. In addition, a latitude-dependent correction factor was applied to correct the latitude bias in the LIS/ODT data, as described in (Lasslop et al., 2014).

    The simulations were initially set up for the study report Aalto and Venäläinen (2021) about current knowledge of the occurrence, monitoring, modelling and suppression of forest fires in Fennoscandia. Simulations were further improved for this

study to match better the observed annual number of fires and burnt areas. Fire duration $D_{\text{fire}}$ depends on population density $P_{\text{D}}$ and fire danger index $FDI = 1 - \frac{\text{fuel moisture}}{\text{moisture of extinction}}$ as follows (Lasslop and Kloster, 2017)

$$D_{\text{fire}} = \begin{cases} \frac{241 \cdot 3}{1 + 240 \cdot e^{-11.06 \cdot FDI}}, P_{\text{D}} \leq 0.01 \\ \frac{241 \cdot (4 - \log(P_D)) \cdot 0.5}{1 + 240 \cdot e^{-11.06 \cdot FDI}}, 0.01 < P_{\text{D}} < 100 \\ \frac{241}{1 + 240 \cdot e^{-11.06 \cdot FDI}}, P_{\text{D}} \geq 100 \end{cases} \tag{1}$$

    The default maximum fire duration (Eq. 1) was reduced from 720 min to 138 min to fit better the reported number of fires and burnt areas (Statistic system of Finnish rescue service database PRONTO, Pelastustoimen resurssi- ja onnettomuustilasto

järjestelmä, PRONTO) in Finland. PRONTO data was available from 2011 to 2018, and wildfires, except fires in fields, grasslands, roadsides and landfills, were selected. The observations were compared to the simulated data for the period 1991–2020. The simulated number of fires was in line with the observed values for Finland, which has a low population density. At high population densities, i.e. in urban areas, the simulations underestimated the number of fires (Fig. A1 a) and burnt areas (Fig. A1 b) compared to observations.

**2.3   Model domain and data analysis**

The monthly or annual means for 9 daily simulate fire variables and 3 derived variables are in dataset Kinnunen (2024). Distributions of fire-related variables were analysed and presented as maps. Additionally, in order to investigate the spatial differences in temporal dynamics in more detail, six example locations around the domain were chosen to be shown in a time series. The selected locations represented different vegetation zones: Location I in the Northern boreal, Location II and III in

the Middle boreal, Location V in the Southern boreal, as well as Location IV and Location VI in the Hemiboreal-Nemoral (Elmhagen et al., 2015). In the selected locations, the fractions of coniferous vegetation were at least 40 % (Fig. 1 a). The reference period 1981–2010 values were calculated as an average of all climate projections. This choice was made because the RCPs and, therefore, climate models only started to deviate from 2006 onwards. In the summer months of June, July and



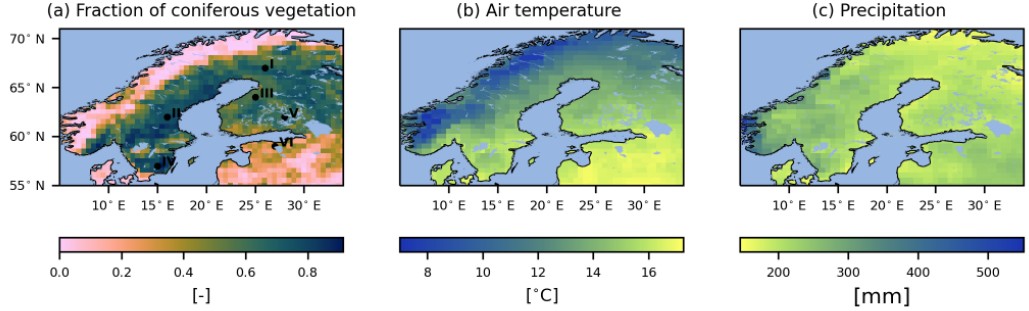

**Figure 1.** (a) The model domain: Fraction of coniferous vegetation and example locations. Location I is in northern Finland (67° N 26° E) 0.728, location II is in central Sweden (62° N 16° E) 0.832, Location III is in central Finland (64° N 25° E) 0.534, Location IV is in southern Sweden (57° N 15° E) 0.833, Location V is in southern Finland (62° N 28° E) 0.566 and Location VI is in Estonia (59° N 27° E) 0.401. (b) Average air temperature [°C] for June, July and August in the reference period 1981–2010 over all the climate projections. (c) Average precipitation [mm] for June, July and August during the reference period 1981–2010 over all all climate projections.

August (JJA), the multi-model average temperature in the domain was 10 °C, increasing from northern to southern and western

to eastern (Fig. 1 b). Multi-model average precipitation for summer months in the domain was 220 mm per year (Fig. 1 c).

The data analysis and plots were done with Python standard functions. The Mann-Kendall trend test ($p \leq 0.05$) was used to test for monotonic trends as implemented in the pyMannKendall package (Hussain and Mahmud, 2019). The colour scales were selected from the scientific colour maps (Crameri et al., 2020).

Yearly averages for the summer months (JJA) or full year were calculated from the daily output for the whole domain.

Variable changes were presented as the difference between averages of the period 2071–2100 and the reference period 1981–2010. The change in monthly climatologies was calculated as a difference between periods 1981–2010 and 2017–2100 in six locations. The relative change was calculated as a ratio of the average of each period, 2071–2100 and 1981–2010. The relative $CO_2$ flux change over time was calculated by comparing the 30-year moving average with the 1981–2010 mean value to smooth out the annual variations and show the overall trend. The time series were created to analyse the trend in the variables,

such as the start and end dates of the fire season. The difference between the average start day and end day of the fire season was calculated to see whether the fire season changed more at the beginning or at the end of the season.

## 3   RESULTS

The largest increase in the summer months (JJA) temperature (ca. 4 °C – 7 °C) is observed under RCP 8.5 CanEMS2 from the period 1981–2010 to 2071–2100 (Fig. A2). The summer temperature increase is larger in the northern study locations,



for example, Location I, than in the southern area, for example, Location IV (Fig. A3). The change in summer precipitation
varied regionally (Fig. A4). The highest precipitation increase (ca. 40 %) is observed under RCP 8.5 CNRM-CM5. The change
in monthly average precipitation from the period 1981–2010 to 2071–2100 varies significantly depending on the climate
projection and the location (Fig. A5). The average gross primary productivity (GPP) during the summer months is greater
during the 2071–2100 period than in the reference period 1981–2010 (Fig. A6).

The litter flux and soil respiration increase, and the amount of fuel decreases for the years 2071–2100 compared to the
reference period 1981–2010 (shown for MIROC5 in Fig. A7). In many areas, projections suggest decreasing the amount of
fuel available for fires because the increase in soil respiration compensates for the increase in litter flux. Typically, the relative
fuel moisture (Fig. A8) is projected to decrease due to an increase in the temperature (Fig. A2). This decrease in moisture leads
to drier and more flammable fuel. The exception is under CNRM-CM5, when fuel moisture is expected to increase. Especially
under RCP 4.5 CNRM-CM5, the fuel appears to be moister in the southern part of the simulated area. The CanESM2 global
climate driver has the highest temperature increase and projects the greatest decrease in the relative fuel moisture.

## 3.1  Fire risk and season

The fire danger index (FDI) indicates the fire risk (Fig. 2) and increases with decreasing fuel moisture (Fig. A8). The FDI
multi-model averaged over the whole domain is 0.2 for the period 1981–2010 (Fig A9 a). Over that period, there are, on
average days of 13 very high or extremely high fire danger (FDI > 0.8) during the summer months (Fig A9 b). All simulations
(except those driven by RCP 4.5 CNRM-CM5) forecast an increase in the probability of spreading fires in Fennoscandia in the
summer months (JJA) from the period 1981–2010 to 2071–2100. The increase in the average FDI is 0.05–0.14 (Fig. 2), and
the increase in the number of very high or extremely high fire danger days is (3.5–12) days (Fig. A10) as the domain average
by the end of the century.

The results for the average changes and their standard deviations (std) of the start day, the end day, and the length of the
fire season of study locations (see Fig. 1 a) are presented in Table 1. The start date change varies from (-6.8 ± 24.7) days to
(-32.4 ± 31.6) days, and the end date from (2.3 ± 23.7) days to (33.2 ± 28.3) days. The lengthening of the fire season varies
from (10.7 ± 34.1) days to (59.1 ± 39.0) days. As a day of the year, the start day of the fire season varies from 88–211 and the
end day from 197–326 in the study locations (Fig. 3). The fire season is assumed to extend by starting at the earliest at the end
of March (Location IV) and ending at the latest in November (Location VI). The trends of fire season start and end dates are
significant according to the Mann-Kendall trend test (p ≤ 0.05), except for the start date under CNRM-CM5 at Locations II
and III and for the end date at Location IV. In the southern part, the fire season is longer, and its lengthening is more extensive
than in the northern part.

In the reference period 1981–2010, the length of the fire season is (87–92) days as, averaged over the model domain. The
length of the fire season is projected to increase by (20–52) days on average in the whole model domain (Fig. A11). The fire
season is estimated to extend (10–23) days at the beginning of the fire season and (10–30) days at the end of the fire season.
When the average change in the start date of the fire season is greater than the change in the end date, the fire season length
increases more at the beginning than at the end of the season. For example, in Location I, the total lengthening of the fire season



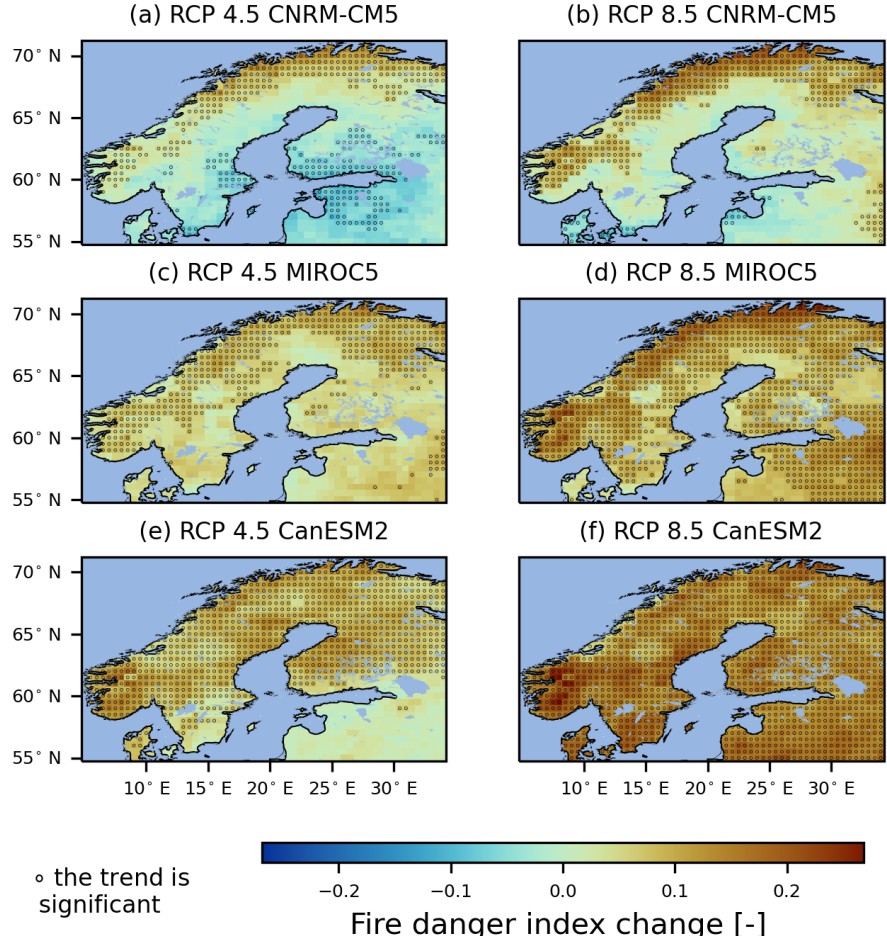

**Figure 2.** Relative changes in the fire danger index (unitless) for the summer months (JJA) average from the reference period 1981–2010 to 2071–2100 under two climate change forcing scenarios and three global driver models. Red indicates an increase, and blue indicates a decrease in the probability of spreading fire from ignition. The dots indicate a significant trend according to the Mann-Kendall test (p ≤ 0.05).

is 23 days under RCP 4.5 CNRM-CM5 and RCP 4.5 MIROC5, but the change is more considerable at the end of the season

in the first case and at the beginning of the season in the second case (Table 1). The lengthening of the fire season is projected to primarily take place at the beginning of the season (max 71 % of grid points) under the CNRM-CM5 and MIROC5 global driver models and in at end of the season (max 79 %) under the CanESM2 global driver model (Fig. A12).



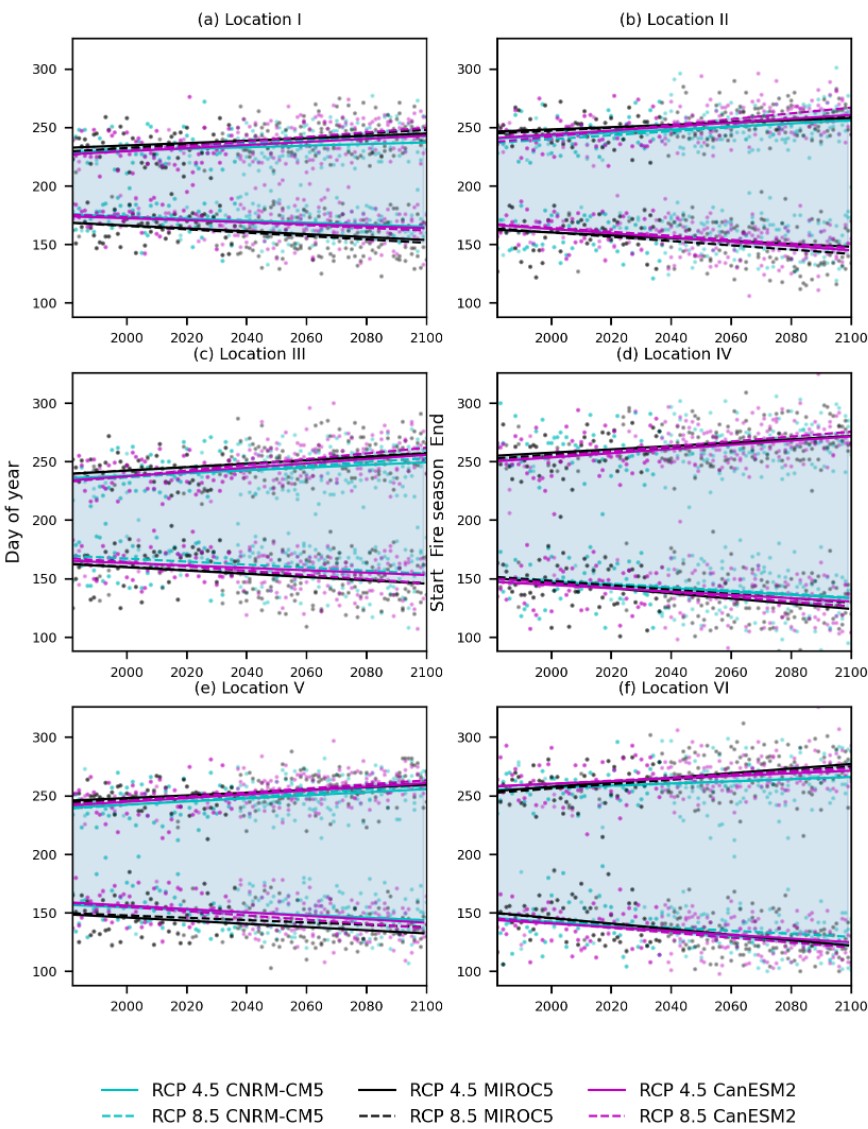

**Figure 3.** Future change in the fire season. Yearly values are dots, and the blue area between trend lines on the start and end dates is the fire season under two climate change forcing scenarios and three global driver models. Trends that are insignificant according to the Mann-Kendall trend test (p ≤ 0.05) are not shown, i.e. the start under CNRM-CM5 at Locations II and III and the end at Location IV. See locations in Fig. 1 a).



## 3.2 Number of fires and burnt area

The simulated human caused ignition rate depends on the population density and has a spatial variation in the range of (0–
0.0012) $km^{-2}yr^{-1}$ (Fig. A13 a). The lightning caused ignition rate has a maximum value around 0.0002 $km^{-2}$ $yr^{-1}$ (Fig. A13
b). Lightning ignition rate is, on average, 7 % of the total ignition rate. The average total ignition rate of 2071–2100 in the
eastern part of the model domain decreases, and in the western part of the model domain, it increases compared to the reference
period 1981–2010 due to the change in population density. The change in the total ignition rate is from -0.0005 $km^{-2}yr^{-1}$ to
0.0006 $km^{-2}yr^{-1}$ (Fig. A13 c).

The number of fires during the reference period (Fig. A9 c) is 0.004 $km^{-2}yr^{-1}$ as a multi-model average over the whole
domain and increases, depending on the model, (0.0006–0.003) $km^{-2}yr^{-1}$ to the end of the century. However, especially in
CNRM-CM5, there are regions of significant decrease by the end of the century (Fig. 4). In Finland, the change in the number
of fires is (-96–1248) fires per year, or (-7–98) % (table 3). The simulations underestimate the average number of fires from
$1355 \pm 509$ to $1568 \pm 556$ in Finland compared to the $1691 \pm 799$ observed fires in PRONTO data (Table 2). The average
number of fires per year is greater in the southern part than in the northern part of Finland, according to both the PRONTO
data and the simulations (Fig. A14). The general pattern is the same, but the simulated values are more evenly distributed and
do not increase as strongly with population density in cities.

The multi-model average burnt area throughout the domain in the reference period is 0.02 $km^2yr^{-1}$ (FigA9). The increase
in the average burnt area by the end of the century is (0.004–0.02) $km^2yr^{-1}$ depending on the model (Fig. 5). Overall, the
changes in the burnt area vary a lot between the model simulations and spatially, even within a single simulation. The greatest
increase of up to around 0.05 $km^2yr^{-1}$, takes place in the southern parts of the domain in RCP 8.5 in CanESM2, while the
largest decrease of -0.02 $km^2yr^{-1}$ is seen in the middle of the domain in RCP 4.5 in CNRM-CM5.

In comparison to the burnt area of PRONTO data (5.84 $km^2$ $\pm$ 3.93 $km^2$), the simulations overestimate the average burnt
area from 7.33 $km^2$ $\pm$ 3.77 $km^2$ to 10.73 $km^2$ $\pm$ 5.86 $km^2$ in Finland (Table 4). According to the simulations, the burnt area
in Finland is estimated to change (-1.52–5.66) $km^2$ or (-18.78–86.64) % from the reference period to the end of the century
(Table 5), resulting in a net burnt area of (6.55–12.20) $km^2$ by the end of the century. The distribution of the average burnt
area in 293 grid points located in Finland shows that the average annual burnt area per grid point is assumed to increase from
the period 1981–2010 to 2071–2100 (Fig. A15). This increase is seen in all simulations except under RCP 4.5 CNRM-CM5.
The amount of emitted $CO_2$ from fires follows the burnt area spatial patterns (Fig. 5). The change of $CO_2$ flux from 2010
to 2100 compared to the reference period (1981–2010) is highly non-linear, indicating either an increase or decrease in $CO_2$
emissions, depending on the climate driver and location (Fig. A16).

## 4 DISCUSSION

Our results indicate a lengthening of the fire season and an increase in the number of fires and the burnt area, even though
the changes in magnitude and even the signs of the changes vary between different driving models and locations. In our
simulations, temperature and precipitation changes are the leading cause of changes in forest fire occurrence in Fennoscandia



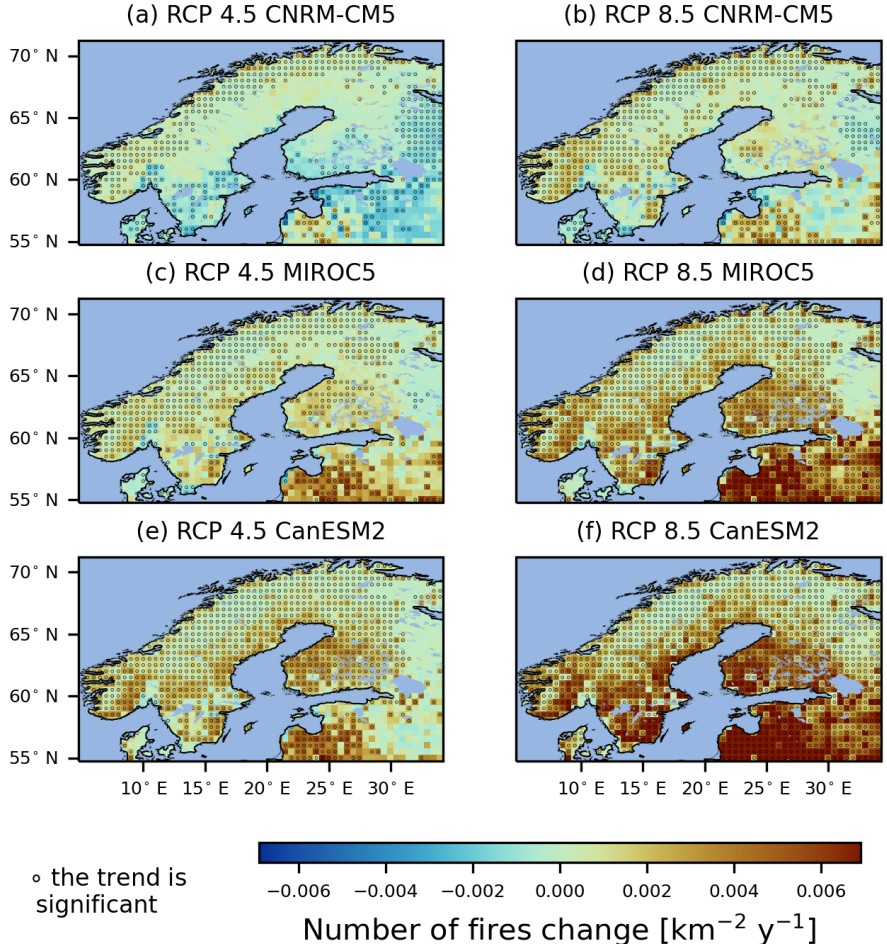

**Figure 4.** The average change in the annual number of fires [km$^{-2}$yr$^{-1}$] from the reference period 1981–2010 to 2071–2100 under two climate change forcing scenarios and three global driver models. The red indicates more fires at the end of the century. The colour bar maximum is limited to show the most important patterns. The dots indicate a significant linear trend according to the Mann-Kendall test (p ≤ 0.05).

and wind effects to the rate of spread. Veira et al. (2016) report that by the end of the century, there can be area-specific changes in forest fire activity due to interactions between climate conditions, population density and land use.

In our simulations, the slight overall decrease in the amount of fuel is a net effect of the increases both in soil respiration and litter flux. Our analyses suggest that in Fennoscandia, fuel availability is not the main limiting factor for fires. The increase in temperature reduces the moisture content of the fuel, making the fuel drier and more flammable. We observed that fuel moisture is the one of the main drivers of the increase in the simulated fire risk. According to Flannigan et al. (2009), the critical elements of fire occurrence and spread are fuel properties, such as type, continuity, structure, heterogeneity, moisture



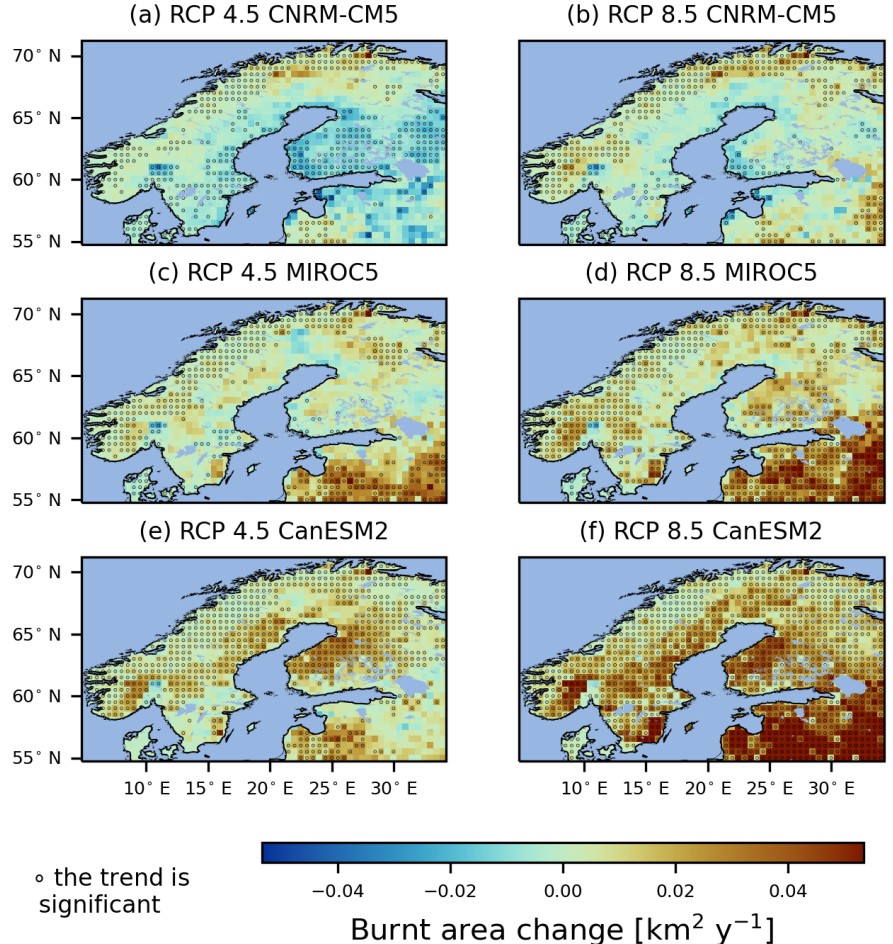

**Figure 5.** Change in the annual average burnt area $[\mathrm{km}^2\mathrm{y}^{-1}]$ from reference period 1981–2010 to 2071–2100 under two climate change forcing scenarios and three global driver models. Red indicates an increase in burnt area. The colour bar maximum is limited to show the most important patterns. The dots indicate a significant linear trend according to the Mann-Kendall test ($p \leq 0.05$).

and volume. The fuel load depends on both the accumulation and decomposition of organic matter, which are affected by climate factors (Kilpeläinen et al., 2010). Moreover, the active management of fuel affects fires (North et al., 2012).

According to our study, the increase in the fire danger index indicates generally a higher probability of spreading fires at the end of the century. Newewless, under CNRM-CM5, we observed decreasing fire risk in the southern parts of the domain. In Finland, Lehtonen et al. (2014) project fire risk to increase by (10–40) % by the end of the century, depending on the GHG scenario. Southern Sweden is projected to have a higher fire risk and northern Sweden to have a lower fire risk than today (Yang et al., 2015; Ramberg, 2020), which is contrary to our results that show an increasing gradient from south to north. Our
simulations indicate an increase of (3.5–12) days in the number of days of high or extremely high fire danger. Mäkelä (2015)



concludes that forest fire danger varies considerably from year to year, and the increase in fire danger days is (7–10) days at the end of the century in Finland. The realised change in the number of fires depends on many factors, but the potential for fires will increase due to changing climatological conditions (Mäkelä, 2015). In our study, the projected increase in the average length of the fire season is (20–52) days. This increase is in line with Veira et al. (2016), who argue that the temperate and boreal fire seasons will become, on average, prolonged by (1–3) months for RCP 8.5 and with Flannigan et al. (2013), who
speculate an increase of more than 20 days per year in the length of the fire season for northern high latitudes. One reason for a longer fire season may be the shortened snow season, especially in southern Finland (Kilpeläinen et al., 2010). The impact of snow cover is not explicitly considered in our simulation. According to our simulations, the lengthening of the fire season is projected to happen typically at the beginning of the season due to warmer and drier weather. In a warming world, fire seasons
should continue to lengthen in temperate and boreal regions (Flannigan et al., 2009).

In our simulations, the probability of lightning was prescribed with a daily climatology that does not include year-to-year variation. The number of days with thunderstorms and the average annual observed cloud-to-ground flash density do not show clear trends in Finland from 1887 to 2018 (Laurila and Mäkelä, 2019). However, Tuomi and Mäkelä (2008) observed large spatial and annual variations in flash density over the 1998–2007 period. The risk of lightning-ignited fires varies from a 62
% decrease to a 38 % increase under RCP 6.0 in the polar regions from the 2010s to the 2090s (Pérez-Invernón et al., 2023). These variations in lightning frequency should be taken into account even though their contribution to the total ignition rate is, on average, only 7 %. Because in Fennoscandia, fires are caused mainly by humans (Mäkelä, 2015; Kilpeläinen et al., 2010) and the ignition rate in simulations is non-linearly dependent on population density, the impact of human ignition should also be further studied with different population density scenarios. Our study demonstrates the potential impacts of climate change
to the fire season, number of fires and burnt area even though JSBACH do not include all relevant aspects of human-nature interaction such as active fire suppression, local landscape fragmentation, land use (e.g. roads) and lakes. Nevertheless, the fire duration limitation serves as a surrogate for fire suppression.

Our simulated annual number of fires in Finland is estimated to be in the range of 1355 ± 509 to 1568 ± 556 between 1991 and 2020, which is lower than the statistical (Finnish rescue service database PRONTO) value of 1691 ± 799. In the SPITFIRE
model, the number of fires is constrained at high population density values, and the statistical value is determined based on emergency reports, encompassing all minor fires. We concluded that the yearly mean simulated burnt area in Finland is from $(7.33 \pm 3.77)$ km$^2$ to $(10.73 \pm 5.86)$ km$^2$ for the period 1991–2020. The observed burnt area through 2011–2018 is lower $(5.84 \pm 3.93)$ km$^2$ due to effective fire detection, management and extinguishing in Finland. As an average over a longer period and broader area, the values should be consistent with the statistical values, even though the simulations are based on scenario
data and do not represent the weather conditions of an actual year. Lehtonen et al. (2016) point out that all over Finland, the fire risk, the number of large fires (>10 ha) and the burnt area are increasing, although due to the current small fire area, one large fire affects the statistics. Larger and more intense fires are expected in a future warmer world (Flannigan et al., 2013). Previous studies show that the SPITFIRE captures the response of a burnt area to precipitation well (Lasslop et al., 2018). There is considerable variation in the fire occurrence between different times and regions due to changes in the natural and
anthropogenic causes impacting the fires (Aakala et al., 2018; Flannigan et al., 2009). The calibration of the SPITFIRE model



may be further improved by the use of observational data sets covering the whole domain and subsequent tuning of model parameters. Changes in human activities and weather conditions cause uncertainty regarding fire risk prediction (Aalto and Venäläinen, 2021).

## 5 Conclusions

In this study, we have studied the projected changes in fire season, number of fires and burnt area over Fennoscandia from the reference period (1981–2010) to the end of the century (2071–2100) using ecosystem model simulations from 1951 to 2100. The simulations suggest increased fire danger due to drier and, thus more flammable fuels towards the end of the century. Increasing soil litter decomposition compensates for the increase in litter input, and less fuel may be available for fires. However, the decrease in fuel is not meaningful enough to limit the occurrence of spreading fires.

Our simulations suggest that the fire season is extended, and the lengthening of fire seasons happens primarily at the beginning of the season. Nevertheless, the spatio-temporal variations in the fire variables depending on global climate driver models (CanESM2, MIROC5 and CNRM-CM5) and regions imply uncertainty in the degree of change. The highest change in temperature leads to an increased fire risk and causes more fires and a larger burnt area, whereas the largest increase in precipitation reduces the fire risk, the number of fires and the burnt area. Moreover, because human activity is the leading cause of fire ignition, our study pinpoints the need for the use of reliable human activity data in addition to improved climate scenario 295 data.

*Data availability.* https://doi.org/10.57707/fmi-b2share.07695381224049c78bd35198d27aaa25

*Author contributions.* OK prepared the manuscript with contributions from all co-authors. LB is responsible for the JSBACH ecosystem 300 model simulations. OK performed the data analysis and produced the graphics and tables with contributions from co-authors. JA and TA have contributed to the design of the study. TM has coordinated the study.

*Competing interests.* The authors declare no competing interests.

*Acknowledgements.* The work is a part of the ACCC Flagship programme (grant No 337552) and funded by the projects 'Forest fires in Fennoscandia under changing climate and forest cover' of the Ministry for Foreign Affairs of Finland IBA funding scheme and 'Evaluating



integrated spatially explicit carbon- neutrality for boreal landscapes and regions' of the Research Council of Finland (grant no. 347860). We
       are thankful to Finnish rescue services for helping us retrieve information from the PRONTO database.



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




**Table 1.** Average changes from the reference period 1981–2010 to 2071–2100 and their standard deviations of the start day, the end day and the length of the fire season in six locations (See Fig. 1) under two climate change forcing scenarios and three global climate driver models.

| Location | Climate projection | Change in the start day (std) [day] | Change in the end day (std) [day] | Change in the length of fire season (std) [day] |
|---|---|---|---|---|
| I<br>67° N 26° E<br>Northern Finland | RCP 4.5 CNRM-CM5 | -9.8 (26.1) | 13.0 (24.6) | 22.8 (33.3) |
| | RCP 4.5 MIROC5 | -13.0 (30.0) | 10.0 (28.0) | 23.0 (43.2) |
| | RCP 4.5 CanESM2 | -8.4 (27.2) | 15.4 (26.9) | 23.8 (37.6) |
| | RCP 8.5 CNRM-CM5 | -12.4 (23.8) | 20.3 (26.5) | 32.6 (36.1) |
| | RCP 8.5 MIROC5 | -17.7 (33.1) | 23.9 (22.6) | 41.6 (37.1) |
| | RCP 8.5 CanESM2 | -14.9 (29.7) | 24.9 (26.9) | 39.8 (43.9) |
| II<br>62° N 16° E<br>Central Sweden | RCP 4.5 CNRM-CM5 | -12.0 (29.1) | 4.6 (29.5) | 16.6 (41.9) |
| | RCP 4.5 MIROC5 | -15.7 (31.9) | 9.8 (22.2) | 25.5 (41.5) |
| | RCP 4.5 CanESM2 | -16.6 (26.2) | 23.0 (29.8) | 39.6 (38.9) |
| | RCP 8.5 CNRM-CM5 | -15.3 (31.2) | 16.9 (28.3) | 32.2 (44.7) |
| | RCP 8.5 MIROC5 | -30.9 (27.7) | 20.9 (23.1) | 51.9 (37.2) |
| | RCP 8.5 CanESM2 | -25.9 (26.5) | 33.2 (28.3) | 59.1 (39.0) |
| III<br>64° N 25° E<br>Central Finland | RCP 4.5 CNRM-CM5 | -15.5 (31.1) | 6.5 (33.4) | 22.0 (49.4) |
| | RCP 4.5 MIROC5 | -20.0 (31.0) | 14.7 (26.7) | 34.8 (43.0) |
| | RCP 4.5 CanESM2 | -12.0 (28.7) | 24.3 (26.3) | 36.3 (36.6) |
| | RCP 8.5 CNRM-CM5 | -22.1 (26.0) | 9.3 (29.5) | 31.4 (41.6) |
| | RCP 8.5 MIROC5 | -26.8 (31.4) | 23.0 (28.4) | 49.8 (46.6) |
| | RCP 8.5 CanESM2 | -22.0 (25.2) | 33.1 (25.7) | 55.2 (33.7) |
| IV<br>57° N 15° E<br>Southern Sweden | RCP 4.5 CNRM-CM5 | -19.5 (37.7) | 2.3 (32.8) | 21.7 (43.6) |
| | RCP 4.5 MIROC5 | -8.4 (32.6) | 2.3 (23.7) | 10.7 (34.1) |
| | RCP 4.5 CanESM2 | -17.0 (29.4) | 24.5 (31.3) | 41.5 (35.7) |
| | RCP 8.5 CNRM-CM5 | -21.8 (39.3) | 17.0 (32.9) | 38.8 (50.6) |
| | RCP 8.5 MIROC5 | -32.4 (31.6) | 19.8 (34.5) | 52.2 (48.3) |
| | RCP 8.5 CanESM2 | -26.2 (28.0) | 23.4 (26.6) | 49.7 (35.8) |
| V<br>62° N 28° E<br>Southern Finland | RCP 4.5 CNRM-CM5 | -7.6 (27.9) | 8.8 (29.9) | 16.4 (41.4) |
| | RCP 4.5 MIROC5 | -6.8 (24.7) | 10.5 (24.4) | 17.3 (36.4) |
| | RCP 4.5 CanESM2 | -9.4 (25.9) | 18.8 (25.1) | 28.1 (39.2) |
| | RCP 8.5 CNRM-CM5 | -7.0 (26.0) | 16.0 (29.1) | 23.0 (45.2) |
| | RCP 8.5 MIROC5 | -19.4 (22.0) | 25.0 (23.7) | 44.4 (33.1) |
| | RCP 8.5 CanESM2 | -21.5 (25.6) | 27.4 (27.0) | 48.8 (35.4) |
| VI<br>59° N 27° E<br>Estonia | RCP 4.5 CNRM-CM5 | -17.7 (32.2) | 7.2 (31.7) | 24.8 (44.9) |
| | RCP 4.5 MIROC5 | -12.8 (30.8) | 11.6 (32.6) | 24.4 (38.2) |
| | RCP 4.5 CanESM2 | -12.3 (27.4) | 11.9 (29.9) | 24.3 (44.8) |
| | RCP 8.5 CNRM-CM5 | -19.1 (34.3) | 9.4 (30.9) | 28.5 (49.5) |
| | RCP 8.5 MIROC5 | -25.0 (29.1) | 18.8 (26.3) | 43.8 (37.8) |
| | RCP 8.5 CanESM2 | -16.5 (25.1) | 21.6 (35.7) | 38.1 (48.5) |





**Table 2.** The average number of fires in Finland during 2011–2018 from PRONTO data and 1991–2020 for simulations.

| Source | Average | Std | Max | Min |
|---|---|---|---|---|
| PRONTO data | 1691 | 799 | 3365 | 652 |
| RCP 4.5 CNRM-CM5 | 1419 | 331 | 2362 | 909 |
| RCP 8.5 CNRM-CM5 | 1414 | 372 | 2363 | 785 |
| RCP 4.5 MIROC5 | 1534 | 448 | 2384 | 710 |
| RCP 8.5 MIROC5 | 1568 | 556 | 3409 | 692 |
| RCP 4.5 CanESM2 | 1355 | 509 | 2655 | 410 |
| RCP 8.5 CanESM2 | 1419 | 436 | 2167 | 429 |

**Table 3.** Average number of fires in Finland and the change by the end of the century.

| Number of fires in Finland | | | |
|---|---|---|---|
| Source | Average 1981–2010 (std) | Average 2071–2100 (std) | Change (%) |
| RCP 4.5 CNRM-CM5 | 1416 (487) | 1320 (487) | -96 (-7) |
| RCP 8.5 CNRM-CM5 | 1386 (488) | 1569 (488) | 183 (13) |
| RCP 4.5 MIROC5 | 1447 (510) | 1807 (510) | 360 (25) |
| RCP 8.5 MIROC5 | 1477 (611) | 2273 (611) | 797 (54) |
| RCP 4.5 CanESM2 | 1253 (503) | 1916 (503) | 663 (53) |
| RCP 8.5 CanESM2 | 1268 (445) | 2516 (445) | 1248 (98) |

**Table 4.** The average burnt areas in Finland during 2011–2018 from PRONTO data and 1991–2020 for simulations.

| Source | Average [$km^2$] | Std [$km^2$] | Max [$km^2$] | Min [$km^2$] |
|---|---|---|---|---|
| PRONTO data | 5.84 | 3.93 | 14.09 | 1.18 |
| RCP 4.5 CNRM-CM5 | 7.68 | 3.10 | 14.93 | 3.01 |
| RCP 8.5 CNRM-CM5 | 7.80 | 3.95 | 17.83 | 2.82 |
| RCP 4.5 MIROC5 | 9.94 | 4.77 | 21.74 | 3.48 |
| RCP 8.5 MIROC5 | 10.73 | 5.86 | 29.3 | 3.70 |
| RCP 4.5 CanESM2 | 7.53 | 4.60 | 19.27 | 0.92 |
| RCP 8.5 CanESM2 | 7.33 | 3.77 | 19.10 | 1.02 |



**Table 5.** The average burnt areas in Finland and the change by the end of the century.

| Source | Average 1981–2010 (std) [km$^2$] | Average 2071–2100 (std) [km$^2$] | Change [km$^2$] (%) |
|---|---|---|---|
| RCP 4.5 CNRM-CM5 | 8.07 (4.27) | 6.55 (4.27) | -1.52 (-18.78) |
| RCP 8.5 CNRM-CM5 | 7.48 (4.27) | 7.64 (4.27) | 0.16 (2.13) |
| RCP 4.5 MIROC5 | 9.48 (5.47) | 10.51 (5.47) | 1.03 (10.90) |
| RCP 8.5 MIROC5 | 9.79 (6.26) | 12.49 (6.26) | 2.70 (27.58) |
| RCP 4.5 CanESM2 | 6.66 (4.22) | 10.45 (4.22) | 3.79 (56.86) |
| RCP 8.5 CanESM2 | 6.54 (3.84) | 12.20 (3.84) | 5.66 (86.64) |

## Appendix A: Figures

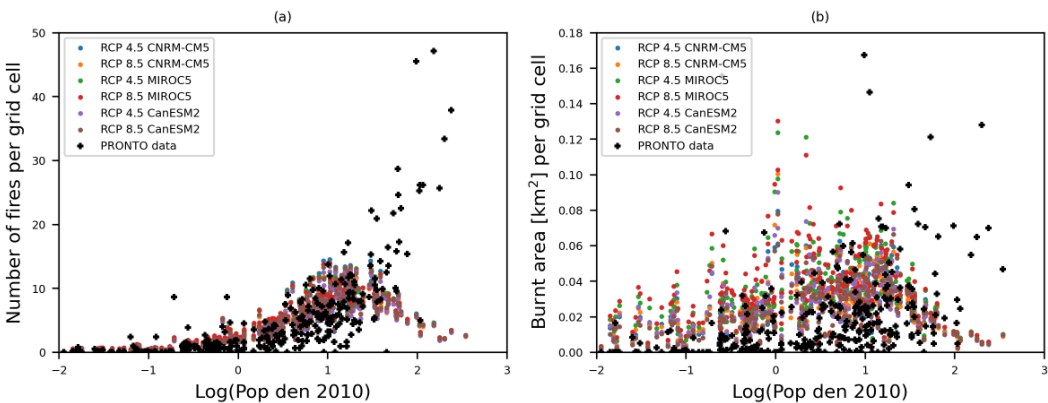

**Figure A1.** Comparison of results from the SPITFIRE model and observations (Finnish rescue service database PRONTO) for Finland. a) Number of fires in Finland per grid cell and b) burnt area [km$^2$] in Finland per grid cell as a function of the logarithm of population density in 2010. Annual averages are calculated for the period 1991–2020 for simulations and 2011–2017 for PRONTO data.

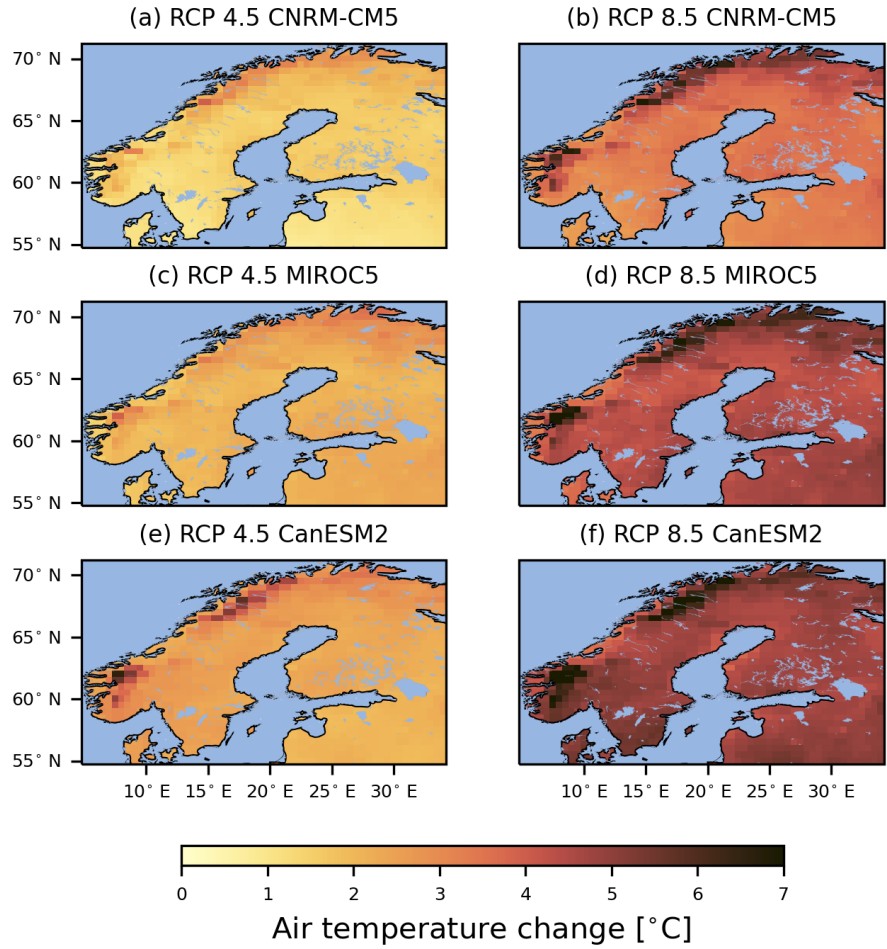

**Figure A2.** Change in the air temperature [°C] during the summer months (JJA) from the period 1981–2010 to 2071–2100 under two climate change forcing scenarios and three climate global driver models. Yellow indicates around a 1-degree increase, and brown indicates around a 6-degree increase.

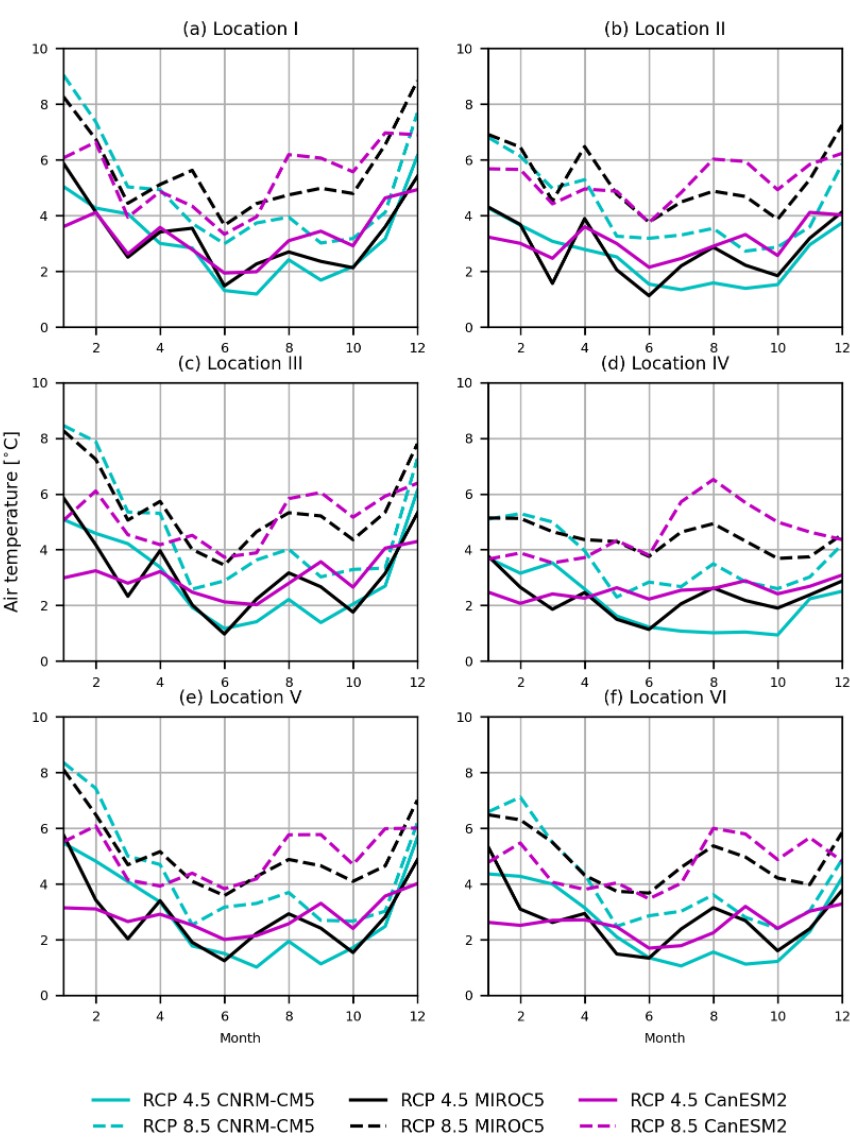

**Figure A3.** Monthly change from the period 1981–2010 to 2071–2100 in average air temperature [°C] at six locations (See locations from Fig. 1). A positive value indicates an increase in temperature.

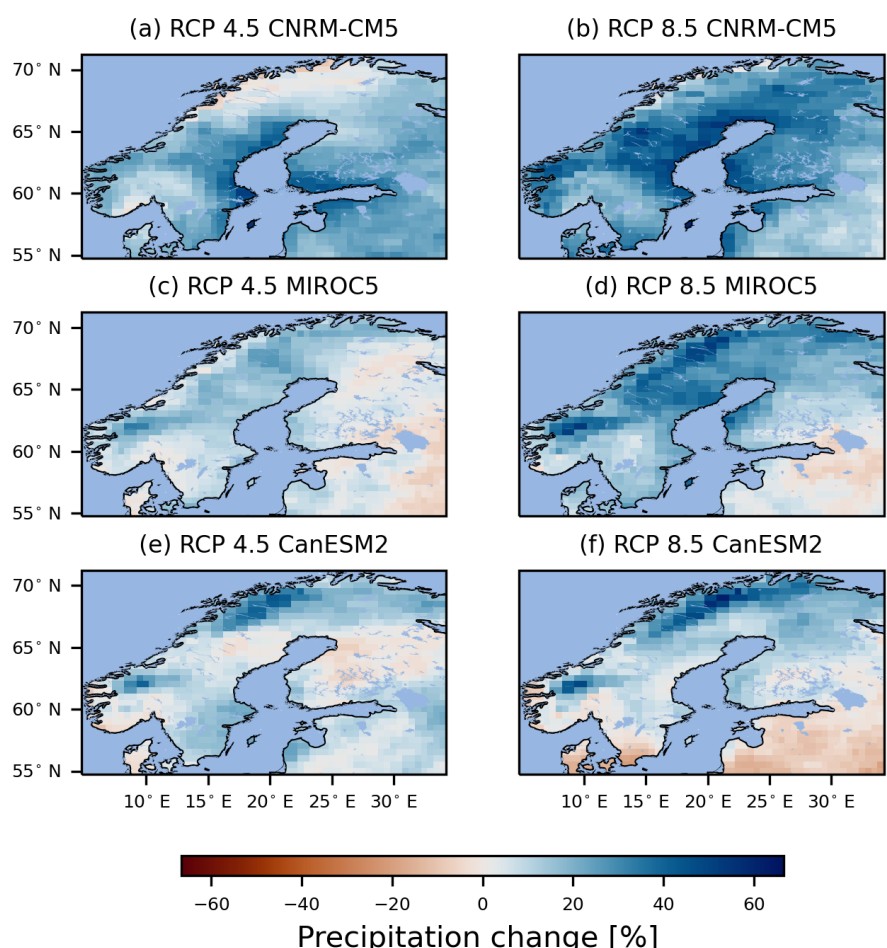

**Figure A4.** The change in the precipitation [%] during the summer months (JJA) from the period 1981–2010 to 2071–2100 under two climate change forcing scenarios and three global climate driver models. The blue (positive) indicates an increase, and the red (negative) indicates decrease.



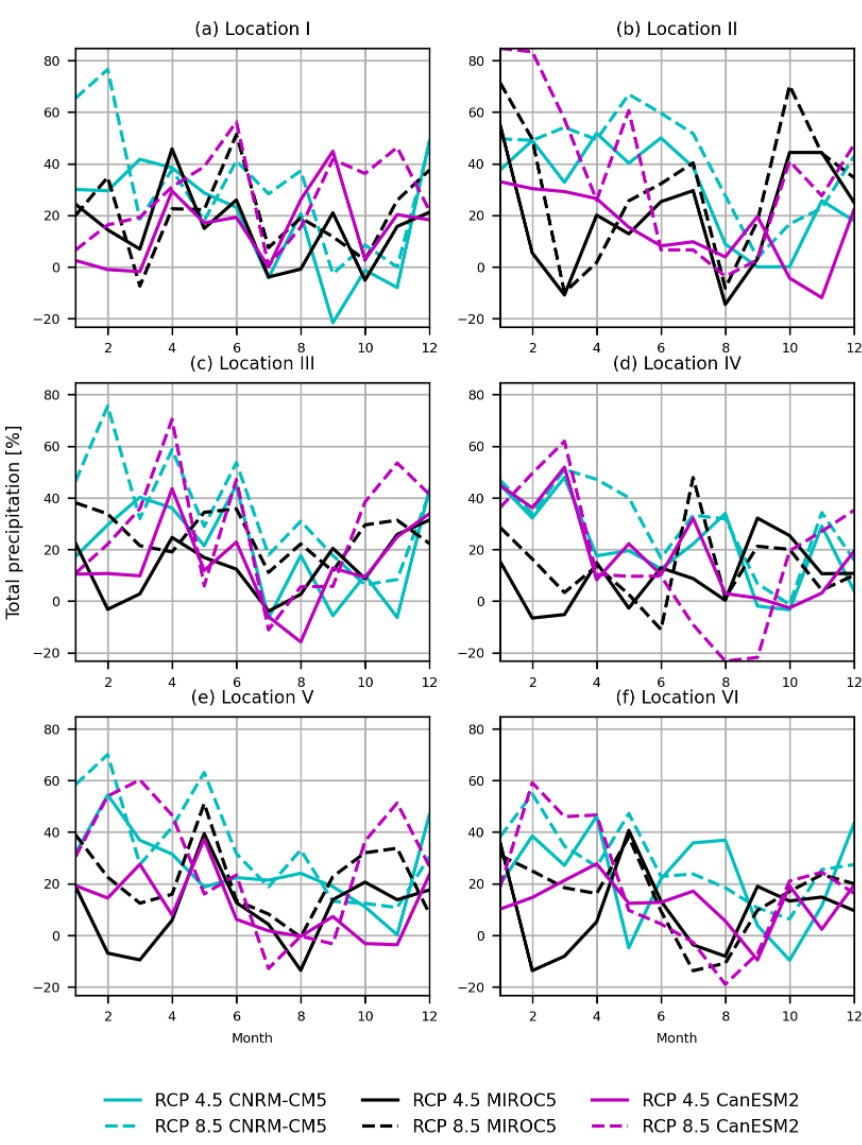

**Figure A5.** Monthly change from the reference period 1981–2010 to 2071–2100 in average precipitation [%] at six locations (See Fig. 1) A positive value indicates an increase in precipitation.



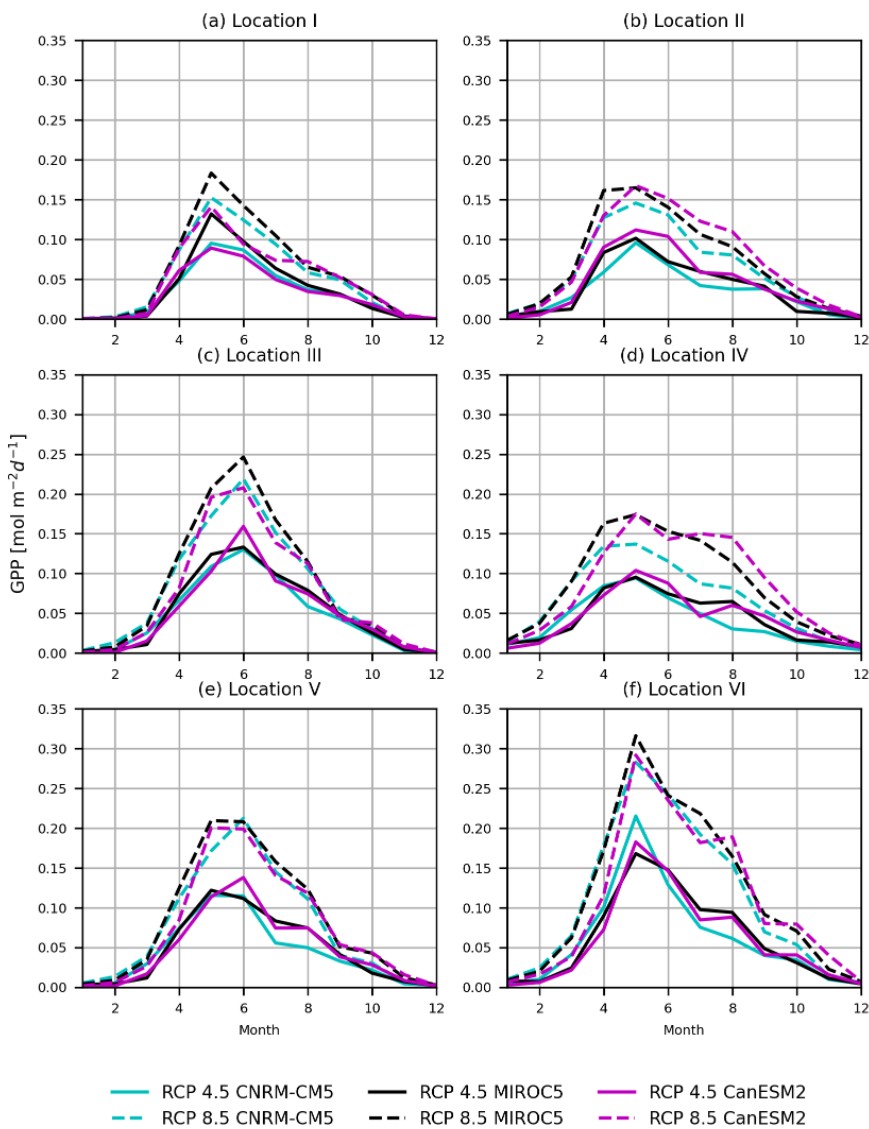

**Figure A6.** Monthly change from the reference period 1981–2010 to 2071–2100 in average gross primary production (GPP) at six locations. (See Fig. 1) The GPP is summed over all the plant functional types.

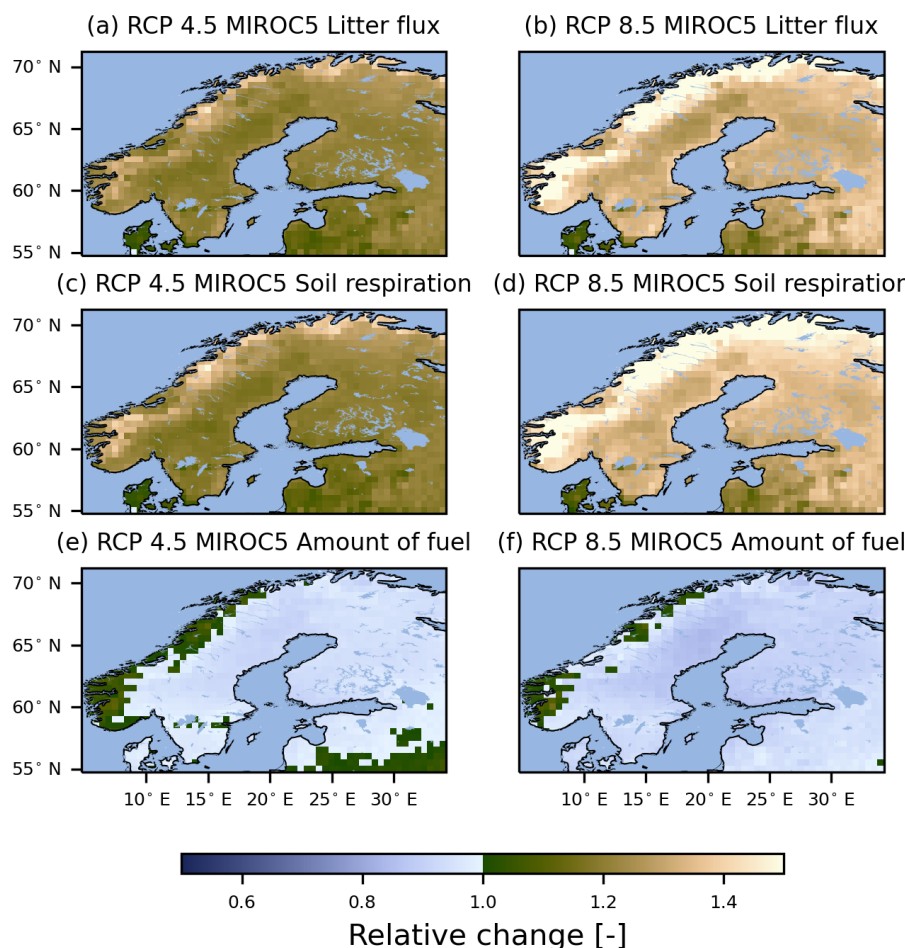

**Figure A7.** Relative change (unitless) in litter flux (a, b), soil respiration (c, d) and amount of fuel (e, f) for the years 2071–2100 compared to the reference period 1981–2010 under two climate change forcing scenarios and the MIROC5 global climate driver model. Less than one (blue) means a decrease, and greater than one (brown) means an increase compared to the reference period.

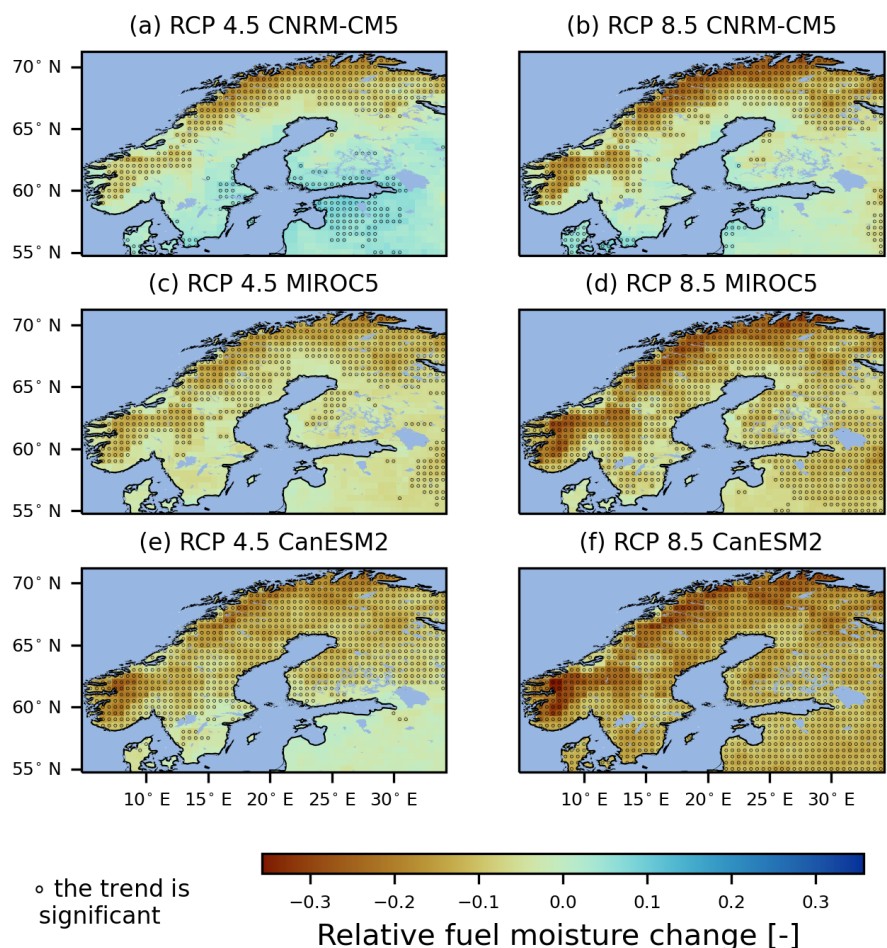

**Figure A8.** Change of average relative fuel moisture (unitless) for the summer months (JJA) from the reference period 1981–2010 to 2071–2100 under two climate change forcing scenarios and three global climate driver models. Red indicates drier fuel in the future. The dots indicate a significant linear trend according to the Mann-Kendall test ($p \leq 0.05$).





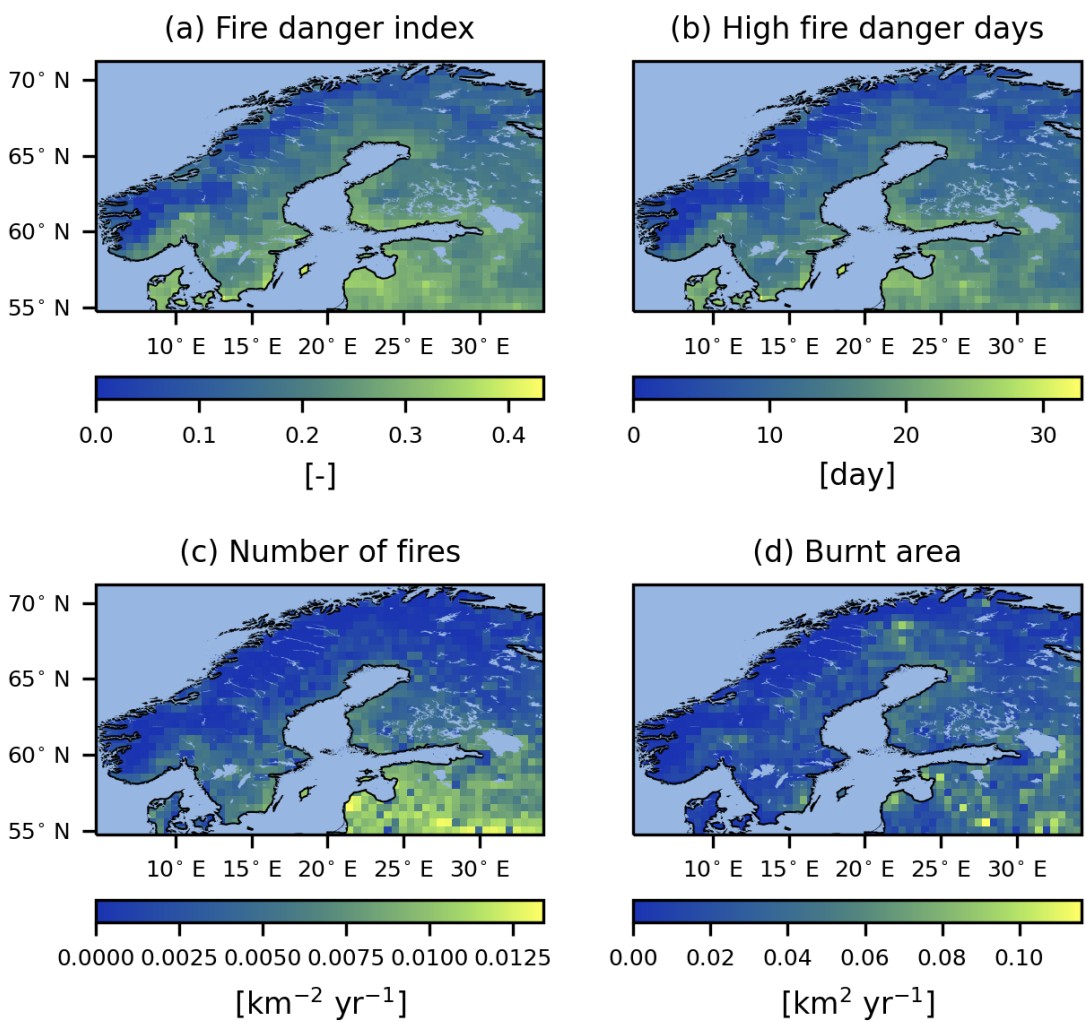

**Figure A9.** Averages over all the climate projections a) fire danger index (unitless), b) number of high fire danger days [day] c) number of fires [$\text{km}^{-2}\text{yr}^{-1}$] and d) burnt area [$\text{km}^2\text{yr}^{-1}$] in the reference period 1981–2010.

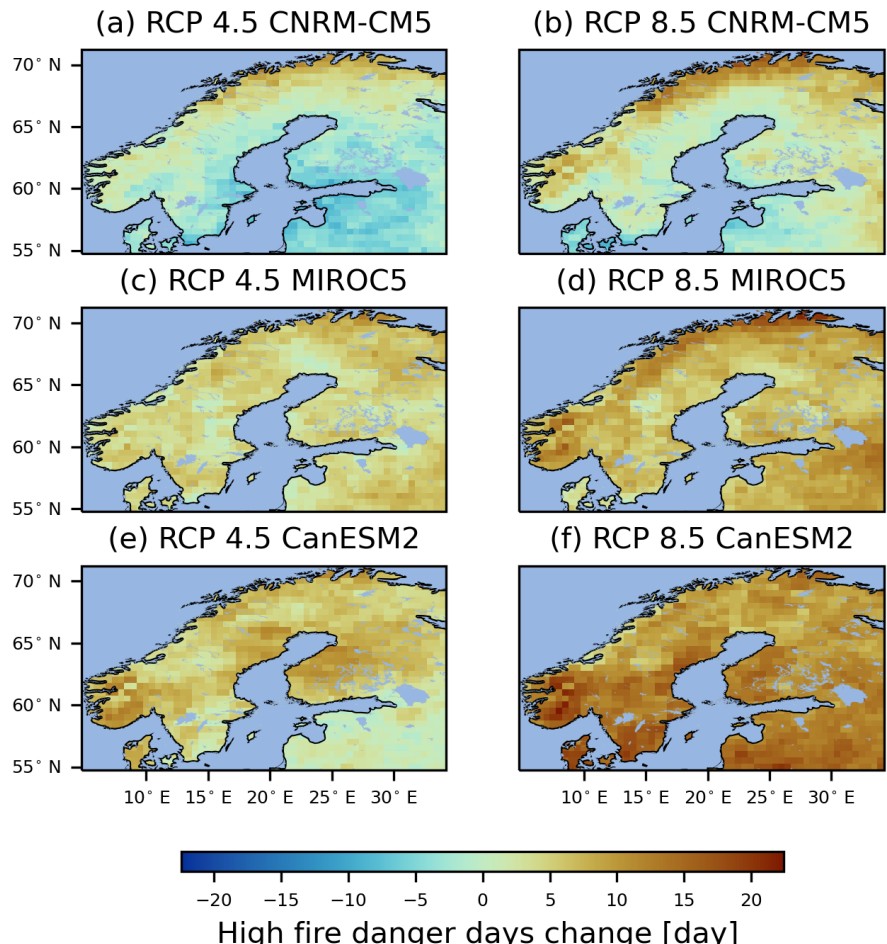

**Figure A10.** Change in the number of very high and extremely high fire danger days [day] from the reference period 1981–2010 to 2071–2100 during the summer months (JJA) under two climate change forcing scenarios and three global climate driver models. Brown indicates an increase in high-fire-risk days.



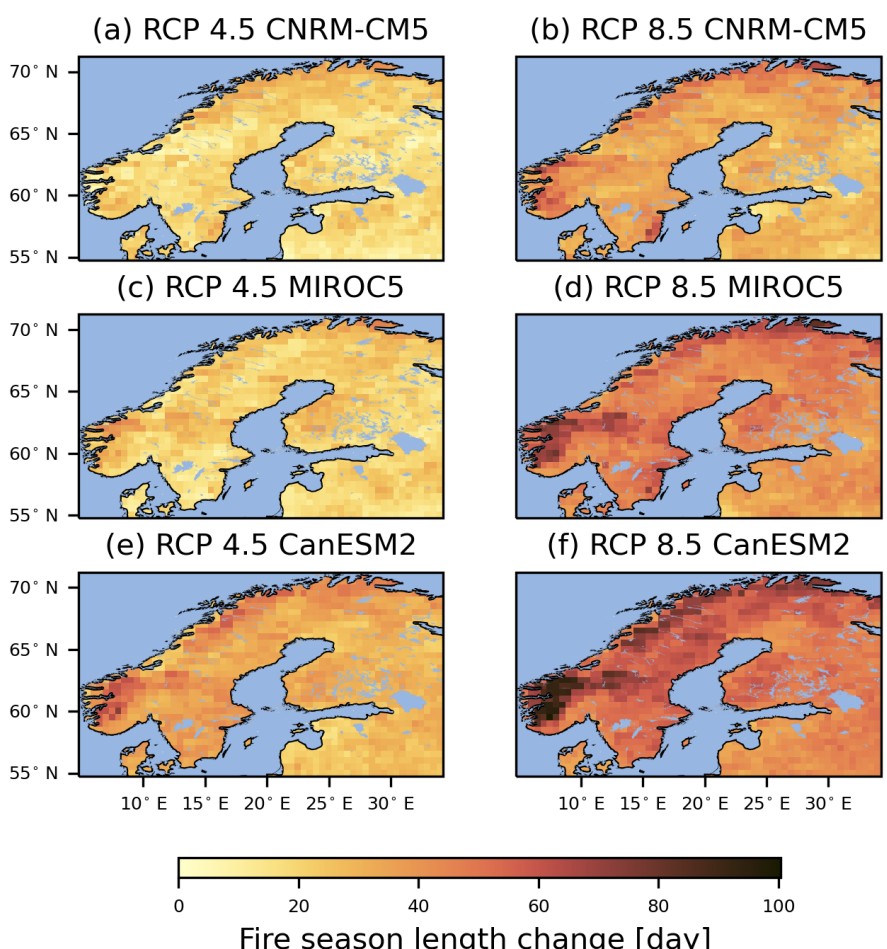

**Figure A11.** Change in the average length of the fire season [day] from the reference period 1981–2010 to 2071–2100 under two climate change forcing scenarios and three global climate driver models.

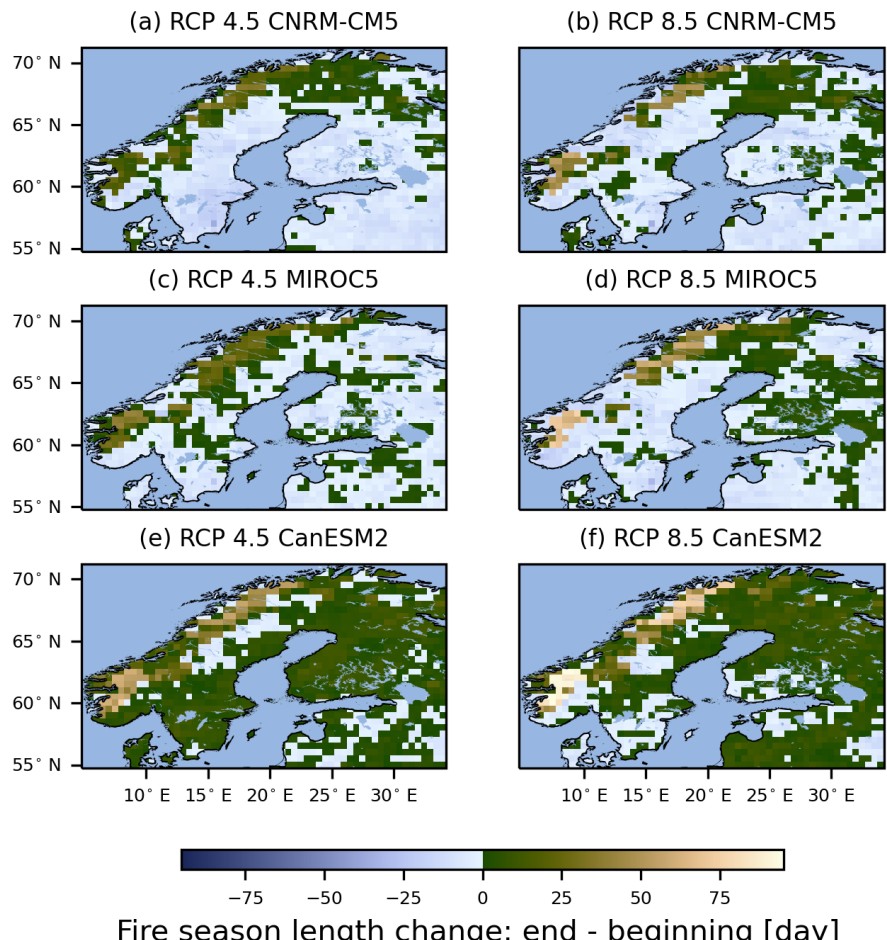

**Figure A12.** The difference between lengthening at the beginning and end of the fire season [day] from the reference period 1981–2010 to 2071–2100 under two climate change forcing scenarios and three global climate driver models. A negative value (blue) indicates that the fire season is lengthening more at the beginning than at the end of the season.



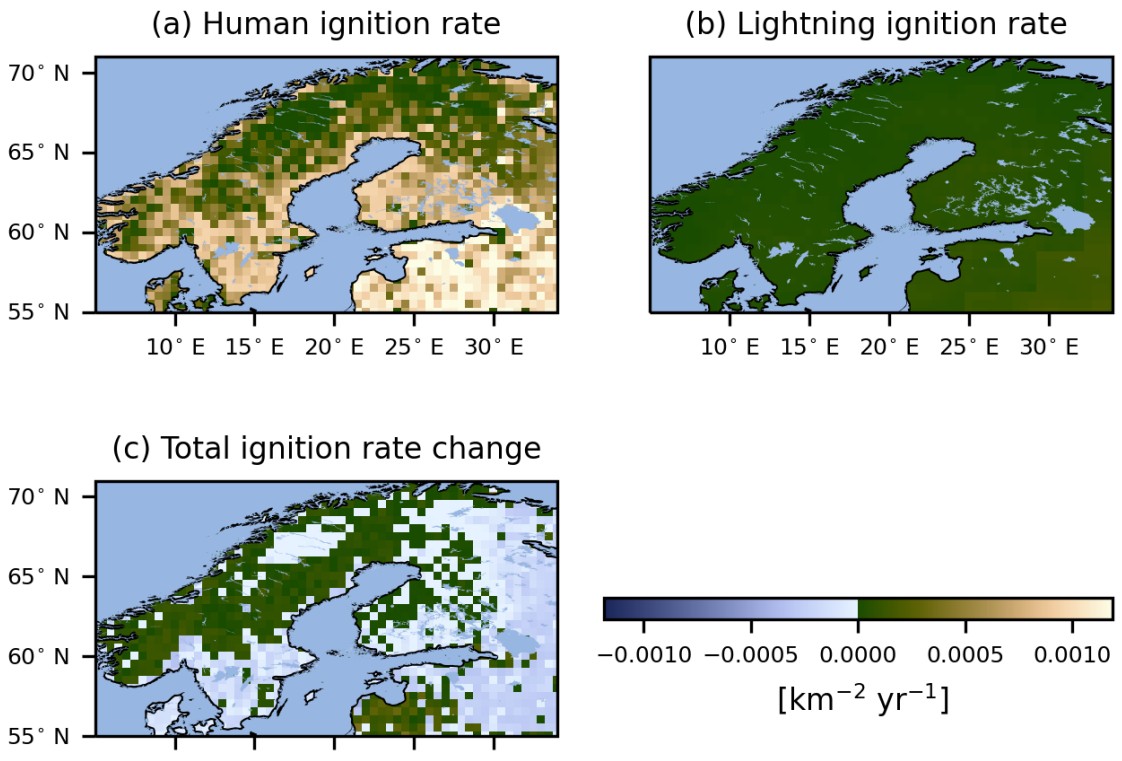

**Figure A13.** Average ignition rate [km$^{-2}$ yr$^{-1}$] caused by a) human or b) lightning in reference period 1981–2010. c) Total ignition rate change from the reference period 1981–2010 to 2071–2100. Light blue indicates less ignitions in the future.



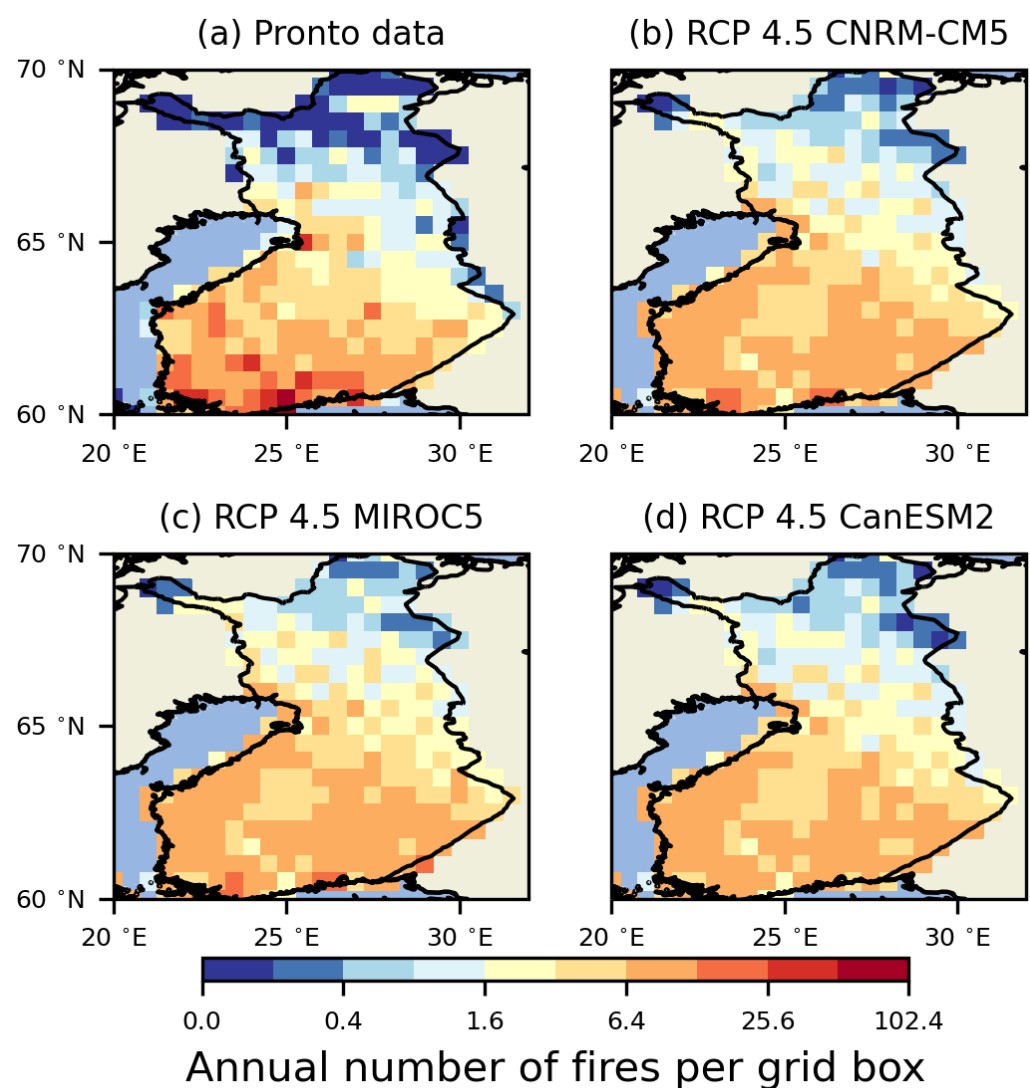

**Figure A14.** The average number of fires per year in Finland by PRONTO data and RCP 4.5 CNRM-CM5, MIROC5 and CanESM2 on a non-linear scale. Yearly means have been calculated at the period 2011–2018 for PRONTO data and 1991–2020 for simulations.



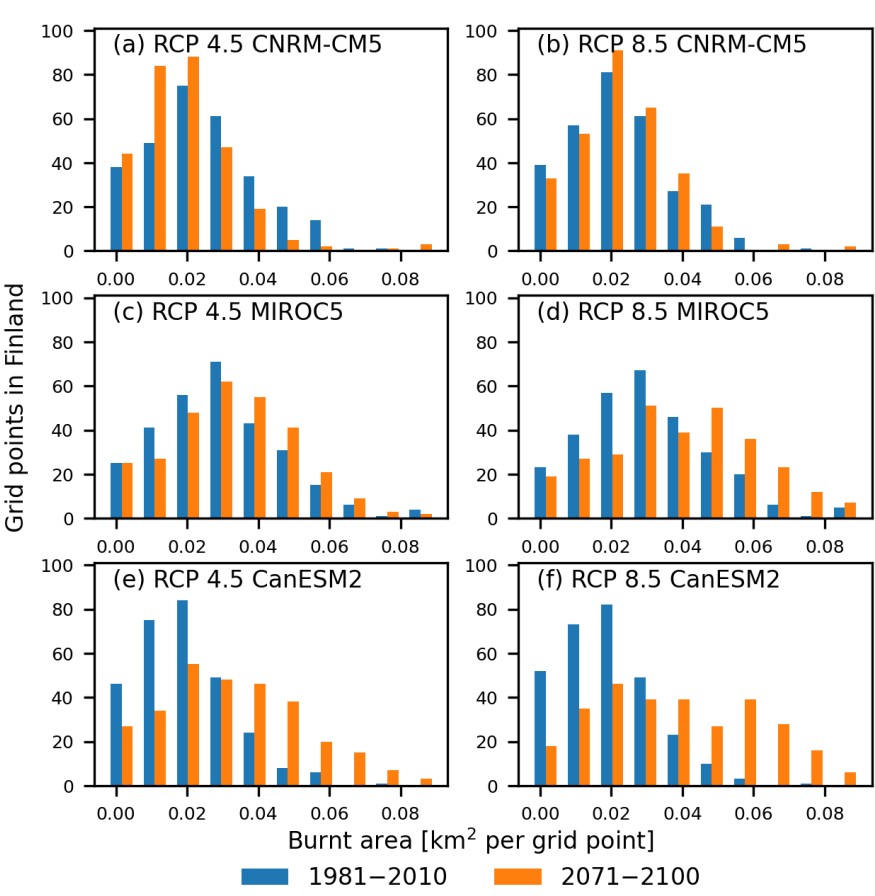

**Figure A15.** The distribution of average burnt areas in 1981–2010 (blue) and 2071–2100 (orange) [km$^2$ per grid point] in Finland under two climate change forcing scenarios and three global climate driver models.



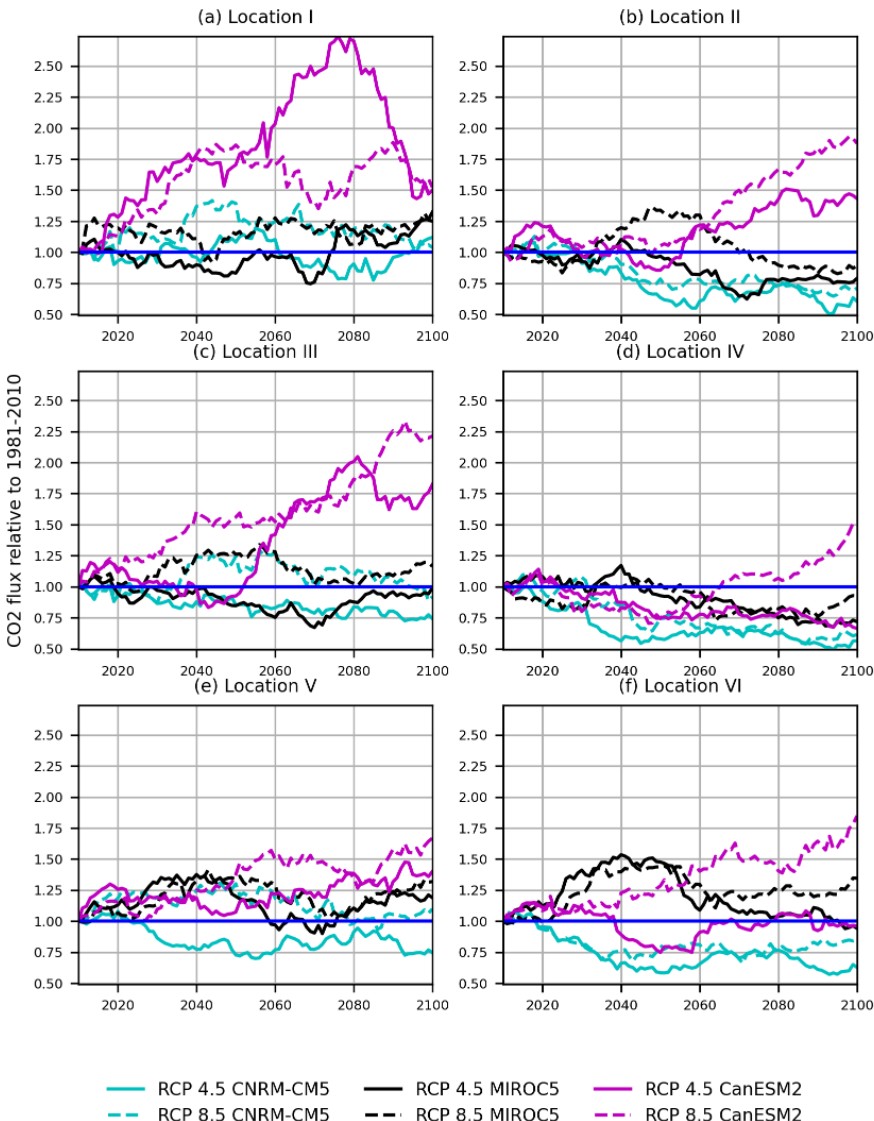

**Figure A16.** Relative change in $CO_2$ flux from fires (unitless) from the reference period 1981–2010 to 2071–2100. In each location (Fig. 1 a), the developments under climate change forcing scenario RCP 4.5 (continuous line) and RCP 8.5 (dashed line) are illustrated with the global climate driver models CNRM-CM5 (cyan), MIROC 5 (black) and CanESM2 (magenta). Above one means an increase, below one means a decrease and one means no change compared to the reference period.