# Peer review of "Projected changes in forest fire season, number of fires and burnt area in Fennoscandia by 2100"

_EGUsphere, 2024_

## Referee Comment (RC2)

Review of "Projected changes in forest fire season, number of fires and burnt area in Fennoscandia by 2100" by Kinnunen et al.

In this paper, Kinnuen et al. projected the length of forest fire season, number of fires and burnt areas in Fennoscandia by 2100, based on simulations from three models that are driven by projected meteorological conditions under climate change scenarios of RCP4.5 and RCP8.5. Overall, the research is well conducted. The results are well articulated and the discussion is appropriate. The topic is suitable and meaningful. The paper is publishable after some revision. I provide the following suggestions for the authors to consider when revising their paper.

It seems that air humidity is overlooked in this research. It is true that relative humidity is related to air temperature; as air temperature increases, relative humidity decreases; this will lead to an increase in fire activities. However, variation of absolute humidity is independent of air temperature. It is reported that trend of fire activities is decreasing in some regions during some of the past decades, possibly due to a wetter atmospheric environment. Is air humidity considered? How?

Please provide more information on FDI, as it is a key variable in this research.

Specific
Abstract: In addition to the range of the variation in the fire variables, the mean values are also important.

L19, "Newer the less"->"Nevertheless".

L19, "point"-> "pointed".

L79, because FDI is a key variable in the simulation, please explicitly express how FDI is estimated. Is there a formula for FDI? What decides environmental dryness, relative humidity, soil moisture or both?

L100, please explain RCP4.5 and RCP8.5 more explicitly. For example, scenarios without additional efforts to constrain emissions ('baseline scenarios') lead to pathways ranging between RCP6.0 and RCP8.5 (IPCC, AR5).

L149, "for the whole domain", is this area-weighted average?

L122, change to "Lasslop et al. (2014)."

L136, which 9 fire variables? Which 3 derived variables?

L136, change to "simulated".

Section 3.1 and 3.2. When discussing the results, the authors provided the mean values and ranges for simulated fire variables, including start and end dates, number of fires, and burnt areas from scenarios. The authors sometimes only provided the ranges for the variables, especially in section 3.2. Please provide the mean values for the simulated results. When doing so, please also explicitly indicate that the mean and associated standard deviation are based on what. Is it an areal mean? Yearly mean? A mean of three models?

L230, "temperature and precipitation are the leading cause of …" How about air humidity?

Conclusions: More quantitative results, in the mean and associated variations, are beneficial to the audience.

---

## Author Comment (AC3)

Dear anonymous referee #2,

Thank you for the opportunity to revise our manuscript, titled "Projected changes in forest fire season, number of fires and burnt area in Fennoscandia by 2100" (egusphere-2024-741). We appreciate the insightful comments and suggestions provided by the reviewers, which have greatly contributed to improving the quality of our work. We have carefully considered all feedback and have made the necessary revisions. Below, we provide a detailed, point-by-point response to each of the reviewers' comments. Our responses are in blue and the line numbers (L) refer to the manuscript. The cited references are provided at the end of the letter.

We hope that these revisions address all the concerns raised by the reviewers. We believe that these changes have strengthened the manuscript, and we look forward to your favorable consideration of our revised submission.

Thank you once again for your time and valuable feedback.

Sincerely,

Outi Kinnunen
Finnish Meteorological Institute
outi.kinnunen@fmi.fi

Review of "Projected changes in forest fire season, number of fires and burnt area in Fennoscandia by 2100" by Kinnunen et al. In this paper, Kinnunen et al. projected the length of forest fire season, number of fires and burnt areas in Fennoscandia by 2100, based on simulations from three models that are driven by projected meteorological conditions under climate change scenarios of RCP4.5 and RCP8.5. Overall, the research is well conducted. The results are well articulated and the discussion is appropriate. The topic is suitable and meaningful. The paper is publishable after some revision. I provide the following suggestions for the authors to consider when revising their paper. It seems that air humidity is overlooked in this research. It is true that relative humidity is related to air temperature; as air temperature increases, relative humidity decreases; this will lead to an increase in fire activities. However, variation of absolute humidity is independent of air temperature. It is reported that trend of fire activities is decreasing in some regions during some of the past decades, possibly due to a wetter atmospheric environment. Is air humidity considered? How? Please provide more information on FDI, as it is a key variable in this research.

We thank you for all your comments and suggestions regarding the article. Thank you for your kind words. We are grateful for your support.

Air humidity is not explicitly used in the model equations. The drying power is estimated using the Nesterov index, which depends on daily maximum temperature, dew point and precipitation. However, the dew point is estimated from the daily minimum temperature. Therefore the fire danger index, FDI, doesn't depend on the atmospheric humidity. Our model shows a decrease in the fire risk with increase in precipitation. We will provide more information on FDI. We hope that these changes will help the reader understand the content better. Details of the changes are presented below.

Specific

Abstract: In addition to the range of the variation in the fire variables, the mean values are also important.

We thank you for the feedback regarding the abstract. We will add the standard deviation for model regional mean, for example (20–52) days to the from (20±7) days to (52±12) days and modify all the respective statistical results accordingly and similarly to the result section.

L19, "Newer the less"->"Nevertheless".

Edited as suggested

L19, "point"-> "pointed".

Edited as suggested

L79, because FDI is a key variable in the simulation, please explicitly express how FDI is estimated. Is there a formula for FDI? What decides environmental dryness, relative humidity, soil moisture or both?

We will add more info about FDI to make it clearer. On line 73 we explain the moisture content of fuel.

We will move the equation of FDI from line 126 to lines 78-79 and number it as equation 1. We will reworde the sentence: The FDI is calculated from environmental dryness, temperature and the availability of fuel (Reick et al., 2021) as follows

FDI = max(0, 1 − fuel moisture/ moisture of extinction)          (1)

We will change the sentence in line 125-126: Fire duration Dfire depends on population density PD and fire danger index FDI (Eq. 1) as follows (Lasslop and Kloster, 2017)

We will change the equation number in line 127 and 128 to 2.

L100, please explain RCP4.5 and RCP8.5 more explicitly. For example, scenarios without additional efforts to constrain emissions ('baseline scenarios') lead to pathways ranging between RCP6.0 and RCP8.5 (IPCC, AR5).

We thank you for the feedback regarding the explaining RCP 4.5 and RCP 8.5.

"The increase of global mean surface temperature by the end of the 21st century (2081–2100) relative to 1986–2005 is likely to be 1.1°C to 2.6°C under RCP 4.5 and 2.6°C to 4.8°C under RCP 8.5. The Arctic region will continue to warm more rapidly than the global mean. " IPCC, 2014

We will add a characterisation of the RCP:s according to these details in the text.

L149, "for the whole domain", is this area-weighted average?

Thank you for this important and helpful comment – it's given us a better understanding of how to make the text clearer. It would be good to be more specific about the context of our calculations. We noticed that the expression 'whole domain' was being used in a different way in the next section. What we meant was that we calculated the yearly averages for each of the grid points we modeled. In the results section, we use the term 'whole domain' when we present the average change across the entire land grid point domain, rather than in a specific location. That's the mean of the values you'll see in the maps. We'll make a few changes to lines 149-152 (in **bold**), the result section (which we'll explain later) and the figure captions (A9, A13 and A15)

*From the daily output values, for every modeled grid point, yearly averages were calculated for the summer months (JJA) of FDI, air temperature, precipitation, fuel moisture and number of very high and extremely high FDI days or for the full year for ignition rates, number of fires and burnt area. Changes in the variables were presented as the difference between averages of the period 2071–2100 and the reference period 1981–2010. We calculated the mean of average changes over the domain in land grid points. The change in monthly climatologies (air temperature, precipitation and GPP) was calculated as a difference between periods 1981–2010 and 2017–2100 in six locations. The relative change was calculated as a ratio of the average of each period, 2071–2100 and 1981–2010 for litter flux, soil respiration and amount of fuel. The relative $CO_2$ flux change over time was calculated by comparing the 30-year moving average with the 1981–2010 mean value to smooth out the annual variations and show the overall trend. The time series were created to analyse the trend in the variables, such as the start and end dates of the fire season. The difference between the average start day and end day of the fire season was calculated to see whether the fire season changed more at the beginning or at the end of the season.*

Figure A9. Averages over all the climate projections **for summer months** a) fire danger index (unitless), b) number of high fire danger days [day] **and for annual** c) number of fires [km−2yr−1] and d) burnt area [km2yr−1] in the reference period 1981–2010.

Figure A13: **Annual** average ignition rate…

Figure A15. The distribution of **annual** average burnt areas in 1981–2010 (blue) and 2071–2100 (orange) [km2 per grid point] in Finland under two climate change forcing scenarios and three global climate driver models

L122, change to "Lasslop et al. (2014)."

According to comments from referee #1, the reference will change to Pierce (1970)

L136, which 9 fire variables? Which 3 derived variables?

We agree that this information should be added. We'll reword the sentence like this: The monthly or annual means for 9 daily simulated fire variables (FDI, number of fires, burnt area, CO2 flux, GPP, litter flux, soil respiration, fuel and fuel moisture) and 3 derived variables (start day, end day and number of very high and extremely high FDI days) are in dataset Kinnunen (2024).

L136, change to "simulated".

Edited as suggested.

Section 3.1 and 3.2. When discussing the results, the authors provided the mean values and ranges for simulated fire variables, including start and end dates, number of fires, and burnt areas from scenarios. The authors sometimes only provided the ranges for the variables, especially in section 3.2. Please provide the mean values for the simulated results. When doing so, please also explicitly indicate that the mean and associated standard deviation are based on what. Is it an areal mean? Yearly mean? A mean of three models?

We apologize for being incoherent. We have now double-checked the values and the text. We noticed that the standard deviation of average 2071-2100 was the same as for period 1981-2010 in tables 3 and 5. We will correct that and add standard deviation for changes. Moreover, we'll change the order of the tables as well as the sentences referring to them respectively. This will improve the readability of the text.

We will add the standard deviation for the presented means. For example (20–52) days to the from (20±7) days to (52±12) days. The averages are calculated for the land grid points, which we try to point out in the text. We will make a few changes to sections 3.1, 3.2 and the abstract.

L230, "temperature and precipitation are the leading cause of …" How about air humidity?

Unfortunately, the fire risk is not explicitly affected by air humidity in the model. The Nesterov index calculations does not include air humidity, but the dew point is estimated from the daily minimum temperature as it is explained in lines 73-76. The drying power is estimated using the NI as follows:

$$\begin{cases} NI = \sum T_{max}(T_{max} - T_{dew}), \text{precipitation} < 3\,mm/day \text{ and } T_{min} - 4 \geqslant 0, \text{when } T_{dew} = T_{min} - 4 \\ NI = 0 \end{cases}$$

Conclusions: More quantitative results, in the mean and associated variations, are beneficial to the audience.

We concidered your comment, however. we decided to not add any quantitative results to the conclusions that they are already presented in section 3, discussed in section 4 and summarised in the abstract.

References:

IPCC, 2014: Climate Change 2014: Synthesis Report. Contribution of Working Groups I, II and III to the Fifth Assessment Report of the Intergovernmental Panel on Climate Change [Core Writing Team, R.K. Pachauri and L.A. Meyer (eds.)]. IPCC, Geneva, Switzerland, 151 pp.

Lasslop, G. and Kloster, S.: Human impact on wildfires varies between regions and with vegetation productivity, Environmental Research Letters, 12, 115 011, https://doi.org/10.1088/1748-9326/aa8c82, 2017

Pierce, E. (1970), Latitudinal Variation of Lightning Parameters, Journal of Applied Meteorology , 9, 194–195.

Reick, C. H., Gayler, V., Goll, D., Hagemann, S., Heidkamp, M., Nabel, J., Raddatz, T., Roeckner, E., Schnur, R., and Wilkenskjeld, S.: JSBACH 3 - The land component of the MPI Earth System Model: Documentation of version 3.2., Berichte zur Erdsystemforschung, doi:10.17617/2.3279802, 2021

---

## Author Response (AR1)

Dear Editor and Reviewers,

We thank the editor and reviewers for their valuable comments and suggestions, which have guided us in refining the manuscript. We have thoughtfully considered each remark and have made the necessary revisions. Our detailed responses to the reviewers' feedback are provided below.

Sincerely,

Outi Kinnunen
Finnish Meteorological Institute
outi.kinnunen@fmi.fi

Dear anonymous referee #1,

Thank you for the opportunity to revise our manuscript, titled "Projected changes in forest fire season, number of fires and burnt area in Fennoscandia by 2100" (egusphere-2024-741). We appreciate the insightful comments and suggestions provided by the reviewers, which have greatly contributed to improving the quality of our work. We have carefully considered all feedback and have made the necessary revisions. Below, we provide a detailed, point-by-point response to each of the reviewers' comments. Our responses are in blue and the line numbers (L) refer to the manuscript. The cited references are provided at the end of the letter.

We hope that these revisions address all the concerns raised by the reviewers. We believe that these changes have strengthened the manuscript, and we look forward to your favorable consideration of our revised submission.

Thank you once again for your time and valuable feedback.

Sincerely,

Outi Kinnunen
Finnish Meteorological Institute
outi.kinnunen@fmi.fi

Review of "Projected changes in forest fire season, number of fires and burnt area in Fennoscandia by 2100" by Kinnunen et al.

This is a land modelling study using JSBACH-SPITFIRE for projected changes to northern European forest fires, driven by bias-corrected climate from 3 ESMs from CMIP5 that have been downscaled to higher resolution (Euro CORDEX). The changes to several fire-related variables, such as the fire number, burned area, fire danger index, and length of fire season were all examined for 30 years near end-of-century, in the context of changing temperature, precipitation, winds, and fuel moisture. They found a large range in the results, depending heavily on the driving ESM, but generally finding an increase in both the number of fires and burned area towards the end of the century in the two RCP scenarios examined (RCP4.5 and RCP8.5). Understanding how wildland fires will change in a changing climate is a very important question, and I recommend this paper be published after the following minor revisions.

We thank the reviewer for the very positive view on the relevance of our work. We hope that these changes, presented below, will help the reader better understand the content.

Minor comments:

Line 19: newer --> never

Edited as suggested.

Line 68: rations --> ratios

Edited as suggested.

Line 90: Can you add a line here that explains if anything is done to account for possibly overlapping fires? E.g. if 2 fires within a grid cell grow large enough, could they merge into one fire that may have less burned area than two distinct fires…?

Overlapping fires are not accounted for. We added sentence to line 92: Overlapping fires were not accounted for, but such occurrences are very rare within the study area.

Line 91-92: "taken into account" how?

We apologize for being unclear. The modeled burning area has been calculated for the forest area in order to make it more comparable with forest fire observations. We reworded the sentence as follows: The analysed burnt area is calculated for forested area.

Line 112: Since land cover changes were not accounted for over the ~150 year period simulated, can you comment on how that would affect your results? Is this region of Europe not expected to have large land cover/vegetation changes? Did it not become more agricultural than forested over time?

Indeed, we did not consider the land cover changes, but used Finnish CORINE and the European CORINE for both past and future. Regarding the past in Finland, the forested area is more than 10 times that of the agricultural fields and the forested area has been relatively constant. There was some decrease in the agricultural land cover between 1950 and 1980, but after that it has been quite stable, especially since ca. 1995. (luke.fi)

Zhou et al. (2021) investigated the recent land cover changes and land transitions from 1992 to 2018 in Norway, Sweden, Finland, and Denmark, as represented by the ESA-CCI-LC and C3S-CDS-LC datasets. According to they findings, the land cover area of forests changes from less than -5% (Sweden) to 4% (Norway) and the changes of the other land cover classes were relatively small as well, compared to the total land area.

Depending on SSP the changes of forested area is forecast to be in between +/-(1%-10%) in most of the Fennoscandia. Denmark and Baltic countries show higher range of variability (Hoffmann et al. 2023). Future land cover changes are out of the scope of this study. Moreover, land cover change does not have strong local legacy effect in the model, because the relatively quickly decomposing carbon in the soil affects the amount of fuel.

We added the following sentences and the references to the end of first paragraph of the Discussion.

"Land use changes were not accounted for. This does not have much impact on the results as according to Zhou et al (2021) forested area have been relatively constant showing changes of the

most prominent land cover class of up to +/-5%. Regarding the future Hoffmann et al. (2023) reported projected land cover changes of up to +/-10% in most of the Fennoscandia."

Lines 117-122: Is a 20% cloud-to-ground fraction applied everywhere in the model domain? If so, do you have a reference for that factor? At higher latitudes, the cloud-to-ground fraction is likely greater than 20%. It could also be a function of cloud-base height.

We thank referee for this insightful comment. The FMI lightning data is cloud to ground, i.e. no need for modification. The LIS/OTD data was modified offline by a latitude dependent factor to obtain cloud to ground lightnings (see below for more details) and used in the south east part of the domain.

Can you comment on how lightning frequency and distribution are expected to change in the RCP4.5 and RCP8.5 scenarios and how this might affect your results (e.g. lines 201-204) in this paper?

This is indeed an interesting point. There is some discussion in the lines 256-260. As far as we understand, there is no consensus on the change. We will add to the Discussion a sentence based on foundings by Rädler et al., who concluded that due to rising instability, in the RCP4.5 and RCP8.5 scenarios, all model members predict relative changes in lightning frequency between 5% and 40% in Northern Europe until 2100.

We added sentence to the line 272: Due to rising atmospheric instability, in the RCP4.5 and RCP8.5 scenarios, all model members predict relative changes in lightning frequency between 5% and 40% in Northern Europe until 2100 (Rädler et al. 2019).

Line 120: ODT --> OTD

Corrected as suggested. Same correction in lines 119 and 122

Line 121: Is the latitude-dependent correction factor applied to the LIS/OTD data a correction to the total lightning flashrate from that dataset? Or do you just mean a latitude-based cloud-to-ground fraction applied to this dataset? If the latter, I wouldn't call it a "correction" since the total flashrate from this dataset is correct. If the former, I can't find this correction in that Lasslop et al (2014) reference. Can you be more clear here about what was done?

The latitude-dependent relation between total flashes and cloud-to-ground flashes is presented in Pierce, 1970 article. It is applied to the LIS/OTD data a correction to the total lightning rate from that dataset.

We added Pierce, 1970 to the references and corrected the sentence: In addition, a latitude-dependent relation between total flashes and cloud-to-ground flashes was applied to correct the latitude bias in the LIS/OTD data (Pierce, 1970; Lasslop et al., 2014).

Line 136: simulate --> simulated

Corrected as suggested.

Figure 1: the 6 locations are difficult to see (black font on dark background colour). Could you please put them on panel (c) instead? Or else, change the font colour to white to better see them.

We changed the font colour to white.

Lines 208-209 (and repeated at lines 268-269): It says the simulations underestimate the number of fires, but then given the uncertainties, those numbers agree with each other. For example, 1355 is within the observed range of 1691 +/- 799, no?

We thank you for the critical comment regarding the conclusions of the number of fires in Finland. We are not able to present accurate limits for the uncertainty because we don't have an estimate of the uncertainty of the annual averages. We only have the average and standard deviation of a few years for the measurements and a few more years of values from the model. We definitely agree that the modeled values are in the range of the measured annual averages. We rewrite the sentence: The simulated values of the average number of fires in Finland from $1355 \pm 509$ to $1568 \pm 556$ match the values $1691 \pm 799$ observed in the PRONTO data (table 2).

Lines 218-219 (and repeated at lines 271-273): Similarly, the burnt area ranges overlap, so I don't think it's correct to say that the simulations overestimate, when they are within the range given (5.84 +/- 3.93 km^2).

We agree and rewrite the sentences as follows: The burnt area in the simulations ranged from 7.33 km2 ± 3.77 km2 to 10.73 km2 ± 5.86 km2 in Finland. Compared to the PRONTO data (5.84 km2 ± 3.93 km2), the simulations slightly overestimate the burnt area, but the dispersion interval covers the averages of the modeled values (Table 4).

Lines 241 & 266: newertheless --> nevertheless

Corrected as suggested.

Lines 259-260: this statement sounds too definitive for being based on one lightning parameterization being implemented in one model (EMAC). Given the high variability and uncertainties associated with both lightning and fire modelling, I suggest you change this sentence to: "The risk of lightning-ignited fires **may vary** from a 62% decrease to a 38% increase under RCP 6.0 in the polar regions from the 2010s to the 2090s**, according to one study** (Pérez-Invernón et al., 2023).

We changed the sentence as suggested. We have added a another sentence and reference in to this context as shown above.

Table 1: this is a big table filled with numbers that is already well-represented by Figure 3. Therefore, I suggest Table 1 be moved to the appendix or supplement.

Answer: We moved the table 1 to the appendix as suggested. The numbering of other tables changed respectively.

Tables 2 & 4, and Figure A14: How come the smaller time frame (2011-2018) was not used for the models too? The different time ranges mean that you are not comparing the same thing from models to the observations. The additional years from the simulations may be responsible for the differences.

As we have used scenarios rather than observed weather data, the model do not represent a real year. We have used an average of 30 years to smooth out the annual variability and gotten more reliable climate than a few years of data. That's why we've always used an average of 30 years. We cannot use the reference period because it does not match the PRONTO data years. Unfortunately, the time period for PRONTO data is short. Below are the values for the shorter period even though

we do not change it in the article. Moreover, we changed the order of the tables as well as the sentences referring to them respectively. This will improve the readability of the text.

Number of fire

| Source 2011-2018 | Average | Std | Max | Min |
|---|---|---|---|---|
| PRONTO data | 1691.0 | 799.0 | 3365.0 | 652.0 |
| RCP 4.5 CNRM-CM5 | 1308.0 | 177.0 | 1640.0 | 1081.0 |
| RCP 8.5 CNRM-CM5 | 1434.0 | 357.0 | 1940.0 | 785.0 |
| RCP 4.5 MIROC5 | 1655.0 | 438.0 | 2384.0 | 1067.0 |
| RCP 8.5 MIROC5 | 1689.0 | 443.0 | 2416.0 | 1002.0 |
| RCP 4.5 CanESM2 | 1417.0 | 385.0 | 2063.0 | 982.0 |
| RCP 8.5 CanESM2 | 1565.0 | 372.0 | 2054.0 | 800.0 |

| Source | Average | Std | Max | Min |
|---|---|---|---|---|
| PRONTO data | 1691 | 799 | 3365 | 652 |
| RCP 4.5 CNRM-CM5 | 1419 | 331 | 2362 | 909 |
| RCP 8.5 CNRM-CM5 | 1414 | 372 | 2363 | 785 |
| RCP 4.5 MIROC5 | 1534 | 448 | 2384 | 710 |
| RCP 8.5 MIROC5 | 1568 | 556 | 3409 | 692 |
| RCP 4.5 CanESM2 | 1355 | 509 | 2655 | 410 |
| RCP 8.5 CanESM2 | 1419 | 436 | 2167 | 429 |

Burnt area

| Source 2011-2018 | Average | Std | Max | Min |
|---|---|---|---|---|
| PRONTO data | 5.84 | 3.93 | 14.09 | 1.18 |
| RCP 4.5 CNRM-CM5 | 6.45 | 1.91 | 9.54 | 4.38 |
| RCP 8.5 CNRM-CM5 | 9.51 | 5.1 | 17.83 | 2.82 |
| RCP 4.5 MIROC5 | 10.07 | 3.92 | 17.83 | 5.4 |
| RCP 8.5 MIROC5 | 12.35 | 4.79 | 19.32 | 4.29 |
| RCP 4.5 CanESM2 | 8.72 | 4.07 | 15.28 | 3.72 |
| RCP 8.5 CanESM2 | 7.96 | 2.53 | 12.16 | 3.77 |

| Source | Average [$km^2$] | Std [$km^2$] | Max [$km^2$] | Min [$km^2$] |
|---|---|---|---|---|
| PRONTO data | 5.84 | 3.93 | 14.09 | 1.18 |
| RCP 4.5 CNRM-CM5 | 7.68 | 3.10 | 14.93 | 3.01 |
| RCP 8.5 CNRM-CM5 | 7.80 | 3.95 | 17.83 | 2.82 |
| RCP 4.5 MIROC5 | 9.94 | 4.77 | 21.74 | 3.48 |
| RCP 8.5 MIROC5 | 10.73 | 5.86 | 29.3 | 3.70 |
| RCP 4.5 CanESM2 | 7.53 | 4.60 | 19.27 | 0.92 |
| RCP 8.5 CanESM2 | 7.33 | 3.77 | 19.10 | 1.02 |

We have added a sentence from the line 132 onwords: For the models, the 30-year average is more reliable than the shorter seven-year PRONTO period, as the forcing data do not represent the conditions in any given year.

Figure A9: It would be helpful if the observations of these same variables were mapped up in a similar way for comparison to these multi-model avg results.

There is no observations of FDI and number of high and extreme high FDI days. We have plotted observations and modeled number of fires in the Figure A14. The plot with burnt area will not give any extra information (shown below).

[Figure]

It makes sense to take the multimodel average over the reference period because the model values are basically the same on average during that period. However, because they differ at the end of the century, by averaging out the models we would miss the information about these differences. The same figure as A9, plotted for the end of the century, is shown above (right hand side). For example, while the number of very high or extremely high fire danger days increases 5.3 as a multimodel average, it changes from 3.5 to 12 days when calculated separately.

We keep the original presentation of the results.

Figure A13: important to note the limitation of panel (c) that future lightning was kept the same as present lightning. Therefore, this panel only shows changes in the future due to the changing human ignition rate. So the title of panel (c) should be "human ignition rate change", no?

We did not change the caption of figure A13, but we reworded the sentence in lines 197-199 as follows: The average total ignition rate of 2071–2100 in the eastern part of the model domain decreases, and in the western part of the model domain, it increases compared to the reference period 1981–2010 due to the change in human ignition rate.

References:

Hoffmann, P., Reinhart, V., Rechid, D., de Noblet-Ducoudré, N., Davin, E. L., Asmus, C., Bechtel, B., Böhner, J., Katragkou, E., and Luyssaert, S.: High-resolution land use and land cover dataset for regional climate modelling: historical and future changes in Europe, Earth Syst. Sci. Data, 15, 3819–3852, https://doi.org/10.5194/essd-15-3819-2023, 2023.

Pierce, E. (1970), Latitudinal Variation of Lightning Parameters, Journal of Applied Meteorology , 9, 194–195.

Rädler, A.T., Groenemeijer, P.H., Faust, E. et al. Frequency of severe thunderstorms across Europe expected to increase in the 21st century due to rising instability. npj Clim Atmos Sci 2, 30 (2019). https://doi.org/10.1038/s41612-019-0083-7

Na Zhou, Xiangping Hu, Ingvild Byskov, Jan Sandstad Næss, Qiaosheng Wu, Wenwu Zhao, Francesco Cherubini (2021), Overview of recent land cover changes, forest harvest areas, and soil erosion trends in Nordic countries, Geography and Sustainability, Volume 2, Issue 3, Pages 163-174, ISSN 2666-6839, https://doi.org/10.1016/j.geosus.2021.07.001. (https://www.sciencedirect.com/science/article/pii/S2666683921000341)

*website luke.fi*

*agricultural land:*
*https://statdb.luke.fi/PxWeb/pxweb/fi/LUKE/LUKE__02%20Maatalous__04%20Tuotanto__22%20Kaytossa%20oleva%20maatalousmaa/03_Peltoala_1910_ja_1920-.px/*
*and forested land:*
*https://statdb.luke.fi/PxWeb/pxweb/fi/LUKE/LUKE__04%20Metsa__06%20Metsavarat/1.01_Metsatalousmaa.px/*

Dear anonymous referee #2,

Thank you for the opportunity to revise our manuscript, titled "Projected changes in forest fire season, number of fires and burnt area in Fennoscandia by 2100" (egusphere-2024-741). We appreciate the insightful comments and suggestions provided by the reviewers, which have greatly contributed to improving the quality of our work. We have carefully considered all feedback and have made the necessary revisions. Below, we provide a detailed, point-by-point response to each of the reviewers' comments. Our responses are in blue and the line numbers (L) refer to the manuscript. The cited references are provided at the end of the letter.

We hope that these revisions address all the concerns raised by the reviewers. We believe that these changes have strengthened the manuscript, and we look forward to your favorable consideration of our revised submission.

Thank you once again for your time and valuable feedback.

Sincerely,

Outi Kinnunen
Finnish Meteorological Institute
outi.kinnunen@fmi.fi

Review of "Projected changes in forest fire season, number of fires and burnt area in Fennoscandia by 2100" by Kinnunen et al. In this paper, Kinnuen et al. projected the length of forest fire season, number of fires and burnt areas in Fennoscandia by 2100, based on simulations from three models that are driven by projected meteorological conditions under climate change scenarios of RCP4.5 and RCP8.5. Overall, the research is well conducted. The results are well articulated and the discussion is appropriate. The topic is suitable and meaningful. The paper is publishable after some revision. I provide the following suggestions for the authors to consider when revising their paper. It seems that air humidity is overlooked in this research. It is true that relative humidity is related to

air temperature; as air temperature increases, relative humidity decreases; this will lead to an increase in fire activities. However, variation of absolute humidity is independent of air temperature. It is reported that trend of fire activities is decreasing in some regions during some of the past decades, possibly due to a wetter atmospheric environment. Is air humidity considered? How? Please provide more information on FDI, as it is a key variable in this research.

We thank you for all your comments and suggestions regarding the article. Thank you for your kind words. We are grateful for your support.

Air humidity is not explicitly used in the model equations. The drying power is estimated using the Nesterov index, which depends on daily maximum temperature, dew point and precipitation. However, the dew point is estimated from the daily minimum temperature. Therefore the fire danger index, FDI, doesn't depend on the atmospheric humidity. Our model shows a decrease in the fire risk with increase in precipitation. We provided more information on FDI. We hope that these changes helps the reader understand the content better. Details of the changes are presented below.

Specific

Abstract: In addition to the range of the variation in the fire variables, the mean values are also important.

We thank you for the feedback regarding the abstract. We added the standard deviation for model regional mean, for example (20–52) days to the from (20±7) days to (52±12) days and modified all the respective statistical results accordingly and similarly to the result and discussion sections.

L19, "Newer the less"->"Nevertheless".

Edited as suggested and added common.

L19, "point"-> "pointed".

Edited as suggested

L79, because FDI is a key variable in the simulation, please explicitly express how FDI is estimated. Is there a formula for FDI? What decides environmental dryness, relative humidity, soil moisture or both?

We added more info about FDI to make it clearer. On line 73 we explain the moisture content of fuel. We moved the equation of FDI from line 126 to lines 78-79 and number it as equation 1. We reworded the sentence: The FDI is calculated from environmental dryness, temperature and the availability of fuel (Reick et al., 2021) as follows

$$FDI = \begin{cases} 1 - \text{fuel moisture/ moisture of extinction}, & \text{fuel moisture/ moisture of extinction} \leq 1 \\ 0, & \text{fuel moisture/ moisture of extinction} > 1 \end{cases}$$

We changed the sentence in line 125-126: Fire duration $D_{fire}$ depends on population density $P_D$ and fire danger index FDI (Eq. 1) as follows (Lasslop and Kloster, 2017)

We changed the equation number in line 127 and 128 to 2.

L100, please explain RCP4.5 and RCP8.5 more explicitly. For example, scenarios without additional efforts to constrain emissions ('baseline scenarios') lead to pathways ranging between RCP6.0 and RCP8.5 (IPCC, AR5).

We thank you for the feedback regarding the explaining RCP 4.5 and RCP 8.5.

"The increase of global mean surface temperature by the end of the 21st century (2081–2100) relative to 1986–2005 is likely to be 1.1°C to 2.6°C under RCP 4.5 and 2.6°C to 4.8°C under RCP 8.5. The Arctic region will continue to warm more rapidly than the global mean. " IPCC, 2014

We added a characterisation of the RCP:s according to these details in the text as follows (line 101): While global mean surface temperature is likely to increase from 1.1°C to 2.6°C under RCP 4.5 and from 2.6°C to 4.8°C under RCP 8.5 by the end of the 21 century relative to 1986-2005 (IPCC, 2014), in Finland the multimodel annual mean temperature increases are 1.9, 3.3. and 5.6 C for RCPs 2.6, 4.5 and 8.5 respectively (Ruosteenoja et al. 2016).

L149, "for the whole domain", is this area-weighted average?

Thank you for this important and helpful comment – it's given us a better understanding of how to make the text clearer. It would be good to be more specific about the context of our calculations. We noticed that the expression 'whole domain' was being used in a different way in the next section. What we meant was that we calculated the yearly averages for each of the grid points we modeled. In the results section, we use the term 'whole domain' when we present the average change across the entire land grid point domain, rather than in a specific location. That's the mean of the values you'll see in the maps. We made a few changes to lines 149-152 (in **bold**), the result section (which we explained later) and the figure captions (A9, A13 and A15)

*For every modeled grid point, multi year averages were calculated for the summer months (JJA) of FDI, air temperature, precipitation, fuel moisture and number of very high and extremely high FDI days. For ignition rates and the annual number of fires and the burnt area multi year averages were calculated. Changes in the variables were presented as the difference between averages of the period 2071–2100 and the reference period 1981–2010. We calculated the mean of the average changes over the domain in land grid points. The change in monthly climatologies (air temperature, precipitation and gross primary productivity) was calculated as a difference between periods 1981–2010 and 2017–2100 in six locations. The relative change was calculated as a ratio of the average of each period, 2071–2100 and 1981–2010 for litter flux, soil respiration and the amount of fuel. The relative $CO_2$ flux change over time was calculated by comparing the 30-year moving average with the 1981–2010 mean value to smooth out the annual variations and show the overall trend. The time series were created to analyse the trend in the variables, such as the start and end dates of the fire season. The difference between the average start day and end day of the fire season was calculated to see whether the fire season changed more at the beginning or at the end of the season.*

Figure A9. Averages over all the climate projections **for summer months** a) fire danger index (unitless), b) number of high fire danger days [day] **and for annual** c) number of fires [km−2yr−1] and d) burnt area [km2yr−1] in the reference period 1981–2010.

Figure A13: **Annual** average ignition rate…

Figure A15. The distribution of **annual** average burnt areas in 1981–2010 (blue) and 2071–2100 (orange) [km2 per grid point] in Finland under two climate change forcing scenarios and three global climate driver models

L122, change to "Lasslop et al. (2014)."

According to comments from referee #1 and editor, we added the reference: Pierce (1970)

L136, which 9 fire variables? Which 3 derived variables?

We agree that this information should be added. We reworded the sentence like this: The monthly or annual means for 9 daily simulated fire variables (FDI, number of fires, burnt area, $CO_2$ flux, gross primary productivity, litter flux, soil respiration, fuel and fuel moisture) and 3 derived variables (start day, end day and number of very high and extremely high FDI days) are in dataset Kinnunen (2024).

L136, change to "simulated".

Edited as suggested.

Section 3.1 and 3.2. When discussing the results, the authors provided the mean values and ranges for simulated fire variables, including start and end dates, number of fires, and burnt areas from scenarios. The authors sometimes only provided the ranges for the variables, especially in section 3.2. Please provide the mean values for the simulated results. When doing so, please also explicitly indicate that the mean and associated standard deviation are based on what. Is it an areal mean? Yearly mean? A mean of three models?

We apologize for being incoherent. We have now double-checked the values and the text. We noticed that the standard deviation of average 2071-2100 was the same as for period 1981-2010 in tables 3 and 5. We corrected that and added standard deviation for changes. Moreover, we changed the order of the tables as well as the sentences referring to them respectively. This will improve the readability of the text.

We added the standard deviation for the presented means. For example (20–52) days to the from (20±7) days to (52±12) days. The averages are calculated for the land grid points, which we try to point out in the text. We made a few changes to sections 3.1, 3.2, discussion and the abstract.

L230, "temperature and precipitation are the leading cause of …" How about air humidity?

Unfortunately, the fire risk is not explicitly affected by air humidity in the model. The Nesterov index calculations does not include air humidity, but the dew point is estimated from the daily minimum temperature as it is explained in lines 73-76. The drying power is estimated using the NI as follows:

$$\begin{cases} \text{NI} = \sum T_{max}(T_{max} - T_{dew}), \text{precipitation} < 3\,\text{mm/day and } T_{min} - 4 \geqslant 0, \text{when } T_{dew} = T_{min} - 4 \\ \text{NI} = 0 \end{cases}$$

Conclusions: More quantitative results, in the mean and associated variations, are beneficial to the audience.

We considered your comment, however. we decided to not add any quantitative results to the conclusions that they are already presented in section 3, discussed in section 4 and summarised in the abstract.

References:

IPCC, 2014: Climate Change 2014: Synthesis Report. Contribution of Working Groups I, II and III to the Fifth Assessment Report of the Intergovernmental Panel on Climate Change [Core Writing Team, R.K. Pachauri and L.A. Meyer (eds.)]. IPCC, Geneva, Switzerland, 151 pp.

Lasslop, G. and Kloster, S.: Human impact on wildfires varies between regions and with vegetation productivity, Environmental Research Letters, 12, 115 011, https://doi.org/10.1088/1748-9326/aa8c82, 2017

Pierce, E. (1970), Latitudinal Variation of Lightning Parameters, Journal of Applied Meteorology , 9, 194–195.

Reick, C. H., Gayler, V., Goll, D., Hagemann, S., Heidkamp, M., Nabel, J., Raddatz, T., Roeckner, E., Schnur, R., and Wilkenskjeld, S.: JSBACH 3 - The land component of the MPI Earth System Model: Documentation of version 3.2., Berichte zur Erdsystemforschung, doi:10.17617/2.3279802, 2021

Ruosteenoja, K., Jylhä, K., and Kämäräinen, M.: Climate projections for Finland under the RCP forcing scenarios., Geophysica, 51, 2016.